# BMP8 and activated brown adipose tissue in human newborns

Adela Urisarri[1,10], Ismael González-García [2,3,10], Ánxela Estévez-Salguero[2,3], María P. Pata[4],
Edward Milbank[2,3], Noemi López [1], Natalia Mandiá[1], Carmen Grijota-Martinez [5], Carlos A. Salgado [6],
Rubén Nogueiras [2,3], Carlos Diéguez[2,3], Francesc Villarroya [3,7], José-Manuel Fernández-Real [3,8,9],
María L. Couce[1✉] & Miguel López [2,3✉]

The classical dogma states that brown adipose tissue (BAT) plays a major role in the regulation of temperature in neonates. However, although BAT has been studied in infants for more than a century, the knowledge about its physiological features at this stage of life is rather limited. This has been mainly due to the lack of appropriate investigation methods, ethically suitable for neonates. Here, we have applied non-invasive infrared thermography (IRT) to investigate neonatal BAT activity. Our data show that BAT temperature correlates with body temperature and that mild cold stimulus promotes BAT activation in newborns. Notably, a single short-term cold stimulus during the first day of life improves the body temperature adaption to a subsequent cold event. Finally, we identify that bone morphogenic protein 8B (BMP8B) is associated with the BAT thermogenic response in neonates. Overall, our data uncover key features of the setup of BAT thermogenesis in newborns.

[1] Neonatology Service, Department of Pediatrics, University Clinical Hospital of Santiago de Compostela, Instituto de Investigación Sanitaria de Santiago de Compostela (IDIS), CIBER Enfermedades Raras (CIBERER), Travesia Choupana, Santiago de Compostela, Spain. [2] Department of Physiology, CIMUS, University of Santiago de Compostela–Instituto de Investigación Sanitaria de Santiago de Compostela (IDIS), Santiago de Compostela, Spain. [3] CIBER Fisiopatología de la Obesidad y Nutrición (CIBERobn), Santiago de Compostela, Spain. [4] Biostatech Advice, Training and Innovation in Biostatistics, S.L. Santiago de Compostela, Spain. [5] Department of Cell Biology, Faculty of Biology, Complutense University, Madrid, Spain. [6] Instituto Galego de Física de Altas Enerxías (IGFAE), University of Santiago de Compostela, Santiago de Compostela, Spain. [7] Departament de Bioquímica i Biomedicina Molecular, Institut de Biomedicina, Universitat de Barcelona (IBUB), and Institut de Recerca Hospital Sant Joan de Déu, Barcelona, Spain. [8] Institut d'Investigació Biomèdica de Girona (IDIBGI) and Department of Medical Sciences, Faculty of Medicine, University of Girona, Girona, Spain. [9] Department of Diabetes, Endocrinology and Nutrition (UDEN), Hospital of Girona Dr. Josep Trueta, Girona, Spain. [10] These authors contributed equally: Adela Urisarri, Ismael González-García.
✉email: Maria.Luz.Couce.Pico@sergas.es; m.lopez@usc.es

Maintaining normothermia at birth is of critical importance and a major challenge for newborn survival. In fact, it is well-established for more than 60 years that neonatal hypothermia is associated with higher mortality, directly or indirectly, as comorbidity associated with infections, preterm birth, or intracranial hemorrhage[1–4]. Despite the maintenance of temperature at this stage of life is essential, and that the prevalence rate of neonatal hypothermia ranges from 32 to 85% in infants born in hospitals[3], the mechanisms by which newborns, when they pass from intra-uterus thermoneutrality to ambient temperature, cope metabolic demands to maintain body temperature, are barely understood. In fact, neonatal infants do not even exhibit the circadian pattern of body temperature seen in older children and adults[5], and the reason for this remains totally unknown.

When ambient temperature falls below thermoneutrality, the instant reaction in mammals, including humans, is to activate heat-saving mechanisms, such as vasoconstriction, piloerection, rounded positions, and reduction of mobility[6,7]. In relation to this, numerous factors make human neonates more susceptible to cold than adults: (i) a higher ratio of body surface area (related to heat loss)/body volume (related to heat production), (ii) a higher proportional surface area of the head, (iii) a lower musculature mass and incapacity to shiver, (iv) lack of thermal insulation, i.e. less subcutaneous fat and fine body hair, and (v) an immature cardiovascular system that does not adapt rapidly and/or properly to cold, for example by regulating vasoconstriction[1,3,4].

Homeotherm (warm-blooded) species have acquired more efficient and long-term mechanisms of non-shivering facultative thermogenesis, in which metabolic mechanisms are activated to produce heat. In mammals, the major site of the facultative thermogenesis is brown adipose tissue (BAT)[7–11]. Brown fat is especially abundant in hibernating mammals and newborns. In human neonates, BAT represents approximately 5% of the body mass and is located along the upper half of the spine and towards the shoulders, mainly surrounding the interscapular area[11–13]. Although BAT has been studied in human fetuses and newborns for more than a century[14–16], the understanding of its physiological features is considerably incomplete. In fact, the classical and long-term assumed dogma is that heat production in brown fat provides alternative means of thermogenesis to neonates[7,8,11,12]. However, apart from morphological and anatomical data, it is completely unknown how brown fat thermogenesis modulates body temperature in human neonates. The main reasons are the ethical constraints (radiation exposure and previous fasting) of performing $^{18}$F fluorodeoxyglucose ($^{18}$F-FDG)-based positron emission tomography–computed tomography (PET-CT) studies in healthy newborns, an approach that has been extensively used to investigate BAT in adult subjects[17–20]. Here, we overcome

these limitations by applying non-invasive infrared thermography (IRT)[21–26] to investigate BAT-induced thermogenesis in healthy human neonates. Thus, the aim of this study has been to investigate the functional link between BAT thermogenesis and the regulation of body temperature in human newborns, as well as the possible association with metabolic and endocrine, correlates modulating neonatal BAT function.

## Results

**Body and BAT temperatures positively correlate in newborns.** A cohort of 50 white newborns was used for this study. Anthropometric parameters [sex, gestational age, body weight, body length, body mass index (BMI, even considering that the use of this parameter is not ideal in human babies due to the low adiposity and the relatively high mass of the head[27]), birth cranial perimeter, age at experimentation and body weight change] from these participants are detailed in Table 1. Firstly, we aimed to investigate the suitability of thermal imaging for the analysis of BAT temperature in newborns. Therefore, we investigated the effect of focus distance from the camera on the thermographic analyses. For that, thermography pictures of newborns at different distances were taken, and analyzed the effects on temperature recordings. In keeping with Planck's radiation law, the radiation of a given body (in this case a newborn) depends on temperature, but not on distance, at least in the range assessed (10–75 cm) (Supplementary Fig. 1a–c). For practical reasons, related to the height of the staff, the position of the babies, and facilities where the images were taken, the focus distance was set to 40–50 cm.

Our data showed that the temperature of the interscapular BAT area of infants exhibited a dynamic and quantifiable range with maximal values recorded around 38 °C (Fig. 1a, the representative image in a 36.5 °C range). Next, we investigated the correlation between BAT temperature and the temperature of other body parts, namely deltoid, neck, and ear outer areas. Notably, when peripheral body temperature was plotted against BAT temperature, a highly significant gender-independent linear correlation was found in all the cases (deltoid, neck, and ear) (Fig. 1b–j). This evidence demonstrates that BAT temperature is associated with body temperature in human newborns. Of note, that correlation was found either in breastfed or formula-fed newborns (Supplementary Fig. 1d, e).

**Cold impacts thermogenesis in newborns.** Next, we wanted to address whether neonatal body and BAT temperatures were affected by cold exposure. The mild cold stimulus consisted in the immersion of the right foot in 19 °C water for 3 min (minute 3 to minute 6), followed by peripheral body and BAT temperature recordings (at 3, 7, and 11 min), as well as blood sampling (Fig. 2a). As a consequence of being born, it is expected for the

## Table 1 Anthropometric parameters of participants.

| | Male | Female | Male and female |
|---|---|---|---|
| N | 28 | 22 | 50 |
| Gestational age at birth (weeks) | 39.7 ± 0.2 | 39.5 ± 0.3 | 39.6 ± 0.2 |
| Birth body weight (g) | 3392.5 ± 72.0 | 3160.0 ± 62.5* | 3290.2 ± 51.0 |
| Birth body length (cm) | 50.3 ± 0.3 | 49.1 ± 0.4* | 49.7 ± 0.3 |
| BMI (kg m$^{-2}$) | 13.4 ± 0.2 | 13.1 ± 0.2 | 13.3 ± 0.1 |
| Birth cranial perimeter (cm) | 35.0 ± 0.2 | 34.2 ± 0.2** | 34.7 ± 0.1 |
| Age day 1 (h) | 11.0 ± 0.8 | 11.9 ± 1.0 | 11.4 ± 0.6 |
| Age day 2 (h) | 33.5 ± 1.1 | 33.8 ± 1.1 | 33.6 ± 0.8 |
| Bodyweight change (age 1–age 2; g) | −163.5 ± 12.6 (N = 23) | −165.2 ± 16.2 (N = 21) | −164.3 ± 10.0 (N = 44) |

Data represent mean ± SEM.
Statistical significance was determined by a two-sided Mann–Whitney test. *P < 0.05, **P < 0.01 vs. males.

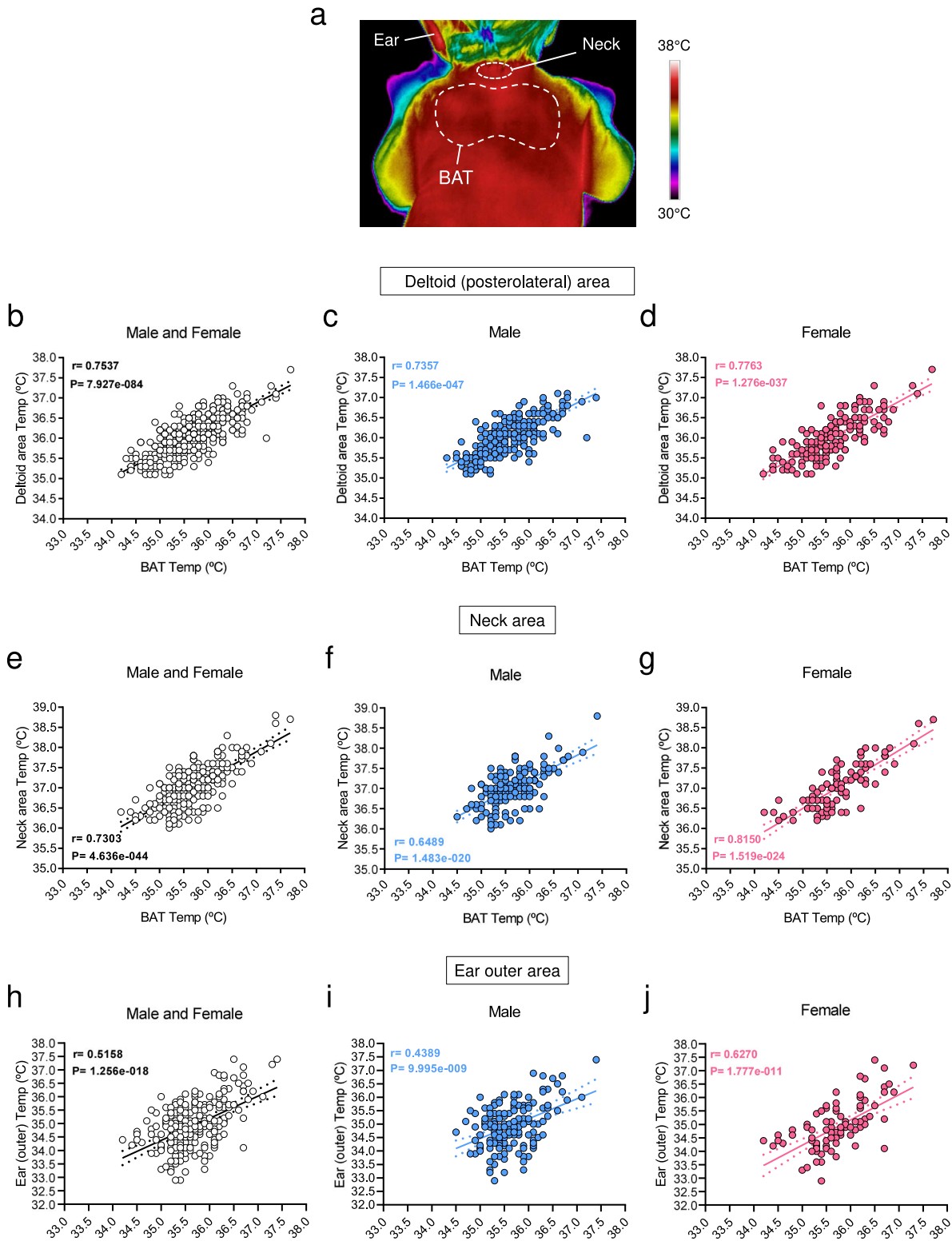

**Fig. 1 Correlations between BAT and body temperature in newborns. a** Representative thermal images and **b–j** correlation between body (deltoid, neck, and ear) and BAT temperatures in newborns. Number of newborns: **b** 449, **c** 270, **d** 179, **e** 255, **f** 158, **g** 97, **h** 249, **i** 156, **j** 93. Association analysis was performed by two-sided Pearson's test; regression line with 95% confidence interval was added when correlation was significant. Source data are provided as a Source data file.

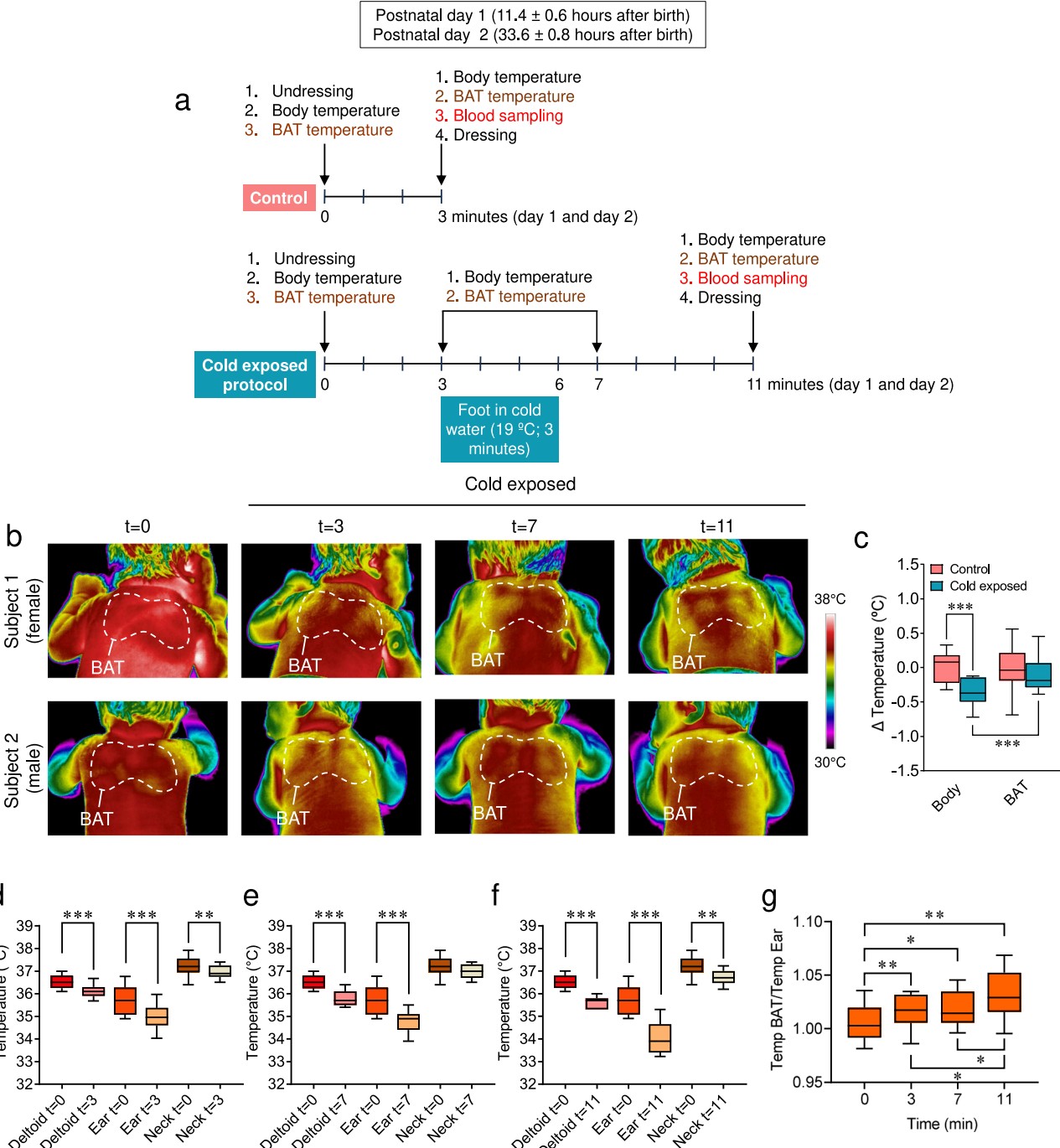

**Fig. 2 Temperature in newborns after the cold stimulus. a** Cold exposure protocol. **b** Representative thermal images of newborns (female upper row and male lower row) at the control and cold exposed conditions: basal (0 min) and 3, 7, and 11 min after the cold stimulus. **c** Body and BAT temperature changes at the control and cold exposed conditions. A number of newborns/group (order of groups): 24, 24, 24, and 25. Box plot indicates median (middle line), 25th, 75th percentile (box), and 10–90th percentiles (whiskers; minima, and maxima, respectively). Body control vs. body cold exposed $P < 0.001$; Body cold exposed vs. BAT cold exposed $P = 0.001$. **d–f** Temperature changes in the deltoid, ear, and neck areas 3, 7, and 11 min after the cold stimulus. Number of newborns/group (order of groups): **d** 33, 32, 33, 33, 32, and 33, **e** 33, 32, 33, 19, 19, and 19, **f** 33, 32, 33, 19, 17, and 18. Box plots indicate median (middle line), 25th, 75th percentile (box), and 10–90th percentiles (whiskers; minima and maxima, respectively). Deltoid $t = 0$ vs. Deltoid $t = 3$, $P < 0.0001$; Ear $t = 0$ vs. Ear $t = 3$ $P < 0.0001$; Neck $t = 0$ vs. Neck $t = 3$ $P = 0.0068$; Deltoid $t = 0$ vs. Deltoid $t = 7$, $P < 0.0001$; Ear $t = 0$ vs. Ear $t = 7$ $P < 0.0001$; Neck $t = 0$ vs. Neck $t = 7$ $P = 0.0888$; Deltoid $t = 0$ vs. Deltoid $t = 11$ $P < 0.0001$; Ear $t = 0$ vs. Ear $t = 11$ $P < 0.0001$; Neck $t = 0$ vs. Neck $t = 11$ $P = 0.017$. **g** Temp BAT/Temp ear ratio at basal condition (0 min) and 3, 7, and 11 min after the cold stimulus. A number of newborns/group (order of groups): 15, 15, 15, and 13. Box plots indicate median (middle line), 25th, 75th percentile (box), and 10–90th percentiles (whiskers; minima and maxima, respectively). $t = 0$ vs. $t = 3$ $P = 0.006$; $t = 0$ vs. $t = 7$ $P = 0.0115$; $t = 0$ vs. $t = 11$ $P = 0.0024$; $t = 3$ vs. $t = 7$ $P = 0.3894$; $t = 3$ vs. $t = 11$ $P = 0.0255$; $t = 7$ vs. $t = 11$ $P = 0.0255$. Statistical significance was determined by two-sided linear mixed models followed by contrast of marginal means differences, with $P$ values adjusted by Holm's method. $*P < 0.05$, $**P < 0.01$, $***P < 0.001$, or $P = 0.001$ for Body cold exposed vs. BAT cold exposed. Statistical parameters for panel **c** are summarized in Supplementary Table 2. Source data are provided as a Source data file.

babies to lose weight in the following days[27]. This is known as neonatal physiological weight loss and ranges between 4% and 7% of the newborn's weight, with a maximum of 10%[27]. It may take them up until 10 days to recover the initial body mass[27]. The loss of body weight is due to different reasons, such as fluid loss and the expulsion of meconium, the earliest stool of a mammalian newborn, including humans[27]. To control the impact of the cold exposure protocol performed in this study on the babies, we initially analyzed the evolution of their body weight during the setting. All the analyzed newborns lost body weight within the physiological range (minimal 1.2% to maximal 8.6%), independently of gender or group and no differences were found between control and cold exposed infants (Supplementary Table 1 and Supplementary Fig. 2a–c). This evidence suggested that the cold challenge did not induce a detectable impact on the newborns' health state.

Our data showed that while a 3-min cold exposure at day 1 induced a marked decrease in peripheral body temperature, recorded in the deltoidal right area ($-0.4 \pm 0.04\,^{\circ}$C; $P < 0.001$; maximal value $-0.2\,^{\circ}$C and minimal value $-1.2\,^{\circ}$C; minute 7), the temperature of the interscapular BAT, located just beside, remained in a similar range in cold and control infants ($-0.08 \pm 0.007\,^{\circ}$C; non-significant BAT control vs. BAT cold exposed; $P < 0.001$ vs. body temp cold exposed) (Fig. 2b, c and Supplementary Table 2). This evidence indicated that the cold stimulus was able to induce a BAT thermogenic response allowing this area to maintain its temperature. Next, we investigated the effect of cold exposure on ear and neck areas temperature (and compared them to the deltoid one), by analyzing the same thermographic images in which BAT temperatures were recorded. Our results showed that cold stimuli elicited a significant decrease in the peripheral temperature in both ear and neck. Of note that action showed a time-dependent pattern, and while the deltoid and ear zones decreased their temperature after 3 and 7 min, the neck region defended much better its temperature and did not show a marked reduction up until 11 min (Fig. 2d–f). The reasons for that effect were likely linked to the anatomical position: the neck region being highly irrigated and anatomically closer to the BAT interscapular area[11–13], whereas the deltoid and particularly the ear being distant. Moreover, the analyzed ear site (the scapha, which is mainly cartilaginous, less irrigated, and subsequently less affected by vasoconstriction) displayed a lower basal temperature and a worse defense after cold exposure at any time (Fig. 2d–f). This evidence was supported by the assessment of the BAT/ear temperature ratio that showed a clear time-dependent activation of the BAT (Fig. 2g).

**Cold impacts circulating factors in newborns.** BAT thermogenesis depends on circulating levels of several hormones which can act directly on brown adipocytes and/or modulate sympathetic tone through hypothalamic centers; among them, thyroid hormones [THs: triiodothyronine (T3) and tetraiodothyronine or thyroxine (T4)], fibroblast growth factor 21 (FGF21), and bone morphogenetic protein 8B (BMP8B) playing a major role[8,28]. Therefore, we aimed to investigate if cold exposure promoted any change in the circulating levels of those factors, as well as glucose and triglycerides (TG), both main fuels used by BAT to enhance its thermogenic activity[7,8]. Our results indicated that whereas the thermogenic stimulus-induced moderate changes in the levels of glucose and TG (Fig. 3a, b and Supplementary Table 3) and did not impact either thyroid hormones (Fig. 3c, d and Supplementary Table 3) or thyroid-stimulating hormone (TSH; Supplementary Fig. 3a), it elicited increases in the levels of FGF21 and BMP8B (Fig. 3e, f and Supplementary Table 3). This suggested that these hormones might influence the thermogenic activity of neonatal BAT.

**Cold correlates thermogenesis and circulating factors in newborns.** To further investigate the possible association between body and BAT temperatures with circulating factors, we performed correlation analyses between these parameters. Our data showed that on basal (non-cold exposed control newborns) conditions at postnatal day 1 ($11.4 \pm 0.6$ h after birth), neither body weight, nor BMI, nor glucose, nor TG or any of the assayed hormones correlated with the body (Fig. 4a, c, e, g, i, k; Supplementary Fig. 4a, c) or BAT temperature (Fig. 4b, d, f, h, l; Supplementary Fig. 4b, d), except for FGF21 (Fig. 4j). Notably, the circulating values of some of the assayed factors, especially T4, T3, and FGF21 showed very wide ranges, which were within the physiological values at that stage of life[27,29–32]. This evidence suggested that BAT thermogenesis was likely not associated with most of those metabolic and endocrine factors at this stage of life and that (i) a longer period of postnatal life or (ii) the presence of thermogenic stimuli might be necessary for the setup of BAT function. To address that, we firstly analyzed the association of both temperatures with circulating factors at postnatal day 2 ($33.6 \pm 0.8$ h after birth). Again, no correlation was found between peripheral body (Fig. 5a, c, e, g, i, k; Supplementary Figs. 4e, g and 5a) or BAT temperature (Figs. 5b, d, f, h, j, l; Supplementary Figs. 4f, h and 5b), either with body weight, BMI, glucose, TG, or any of the hormones assayed. From this data, it could be concluded that although timing increased the circulating levels of FGF21 and BMP8B (Fig. 3e, f), in the absence of thermogenic stimuli only FGF21 levels were associated with BAT temperature in neonates.

Next, we investigated the effect of cold exposure on the correlations between temperatures (body and BAT), body weight, BMI, circulating glucose, TG, and hormones. The analysis at day 1, after 3 min of thermogenic stimulation, did not reveal any association/correlation between body/BAT temperatures and circulating factors (glucose vs. body temperature $r = 0.08121$, $P = 0.6933$; glucose vs. BAT temperature $r = 0.2358$, $P = 0.2565$; TG vs. body temperature $r = 0.3661$, $P = 0.09376$; TG vs. BAT temperature $r = 0.02495$, $P = 0.9145$; T4 vs. body temperature $r = 0.2618$, $P = 0.1964$; T4 vs. BAT temperature $r = 0.1169$, $P = 0.5780$; T3 vs. body temperature $r = 0.1433$, $P = 0.4850$; T3 vs. BAT temperature $r = 0.1303$, $P = 0.5349$; FGF21 vs. body temperature $r = 0.3475$, $P = 0.1577$; FGF21 vs. BAT temperature $r = 0.3732$, $P = 0.1271$; BMP8B vs. body temperature $r = 0.1873$, $P = 0.5038$; BMP8B vs. BAT temperature $r = 0.3082$, $P = 0.2836$). This lack of correlation was, in any case, reasonable, as the endocrine thermogenic response did not have enough time (3 min of cold exposure) to be completed and exert a functional effect. Therefore, we analyzed the possible associations in cold exposed infants at postnatal day 2, after being exposed to cold at day 1 following a similar protocol. Notably, our analyses identified significant correlations between circulating levels of glucose and T4 with body temperature (Fig. 6a, e). No associations were found between either BAT temperature and glucose or T4 (Fig. 6b, f) or any of the temperatures with body weight, BMI, TG, T3, FGF21, or TSH (Fig. 6c, d, g, h–j; Supplementary Figs. 4i–l and 5c, d). Notably, serum BMP8B showed a highly significant correlation with both body and BAT temperatures (Fig. 6k, l). Overall, these data indicated that cold exposure, and not postnatal age, was the key factor inducing hormonal thermogenic responses in human newborns.

**Cold improves following thermogenic responses in newborns.** Finally, we aimed to address whether the thermogenic and hormonal responses elicited by a mild cold stimulus might promote a better thermogenic reaction in subsequent exposure events. If so, this would imply that neonates would defend their body temperature more efficiently after the immersion of their foot in cold water

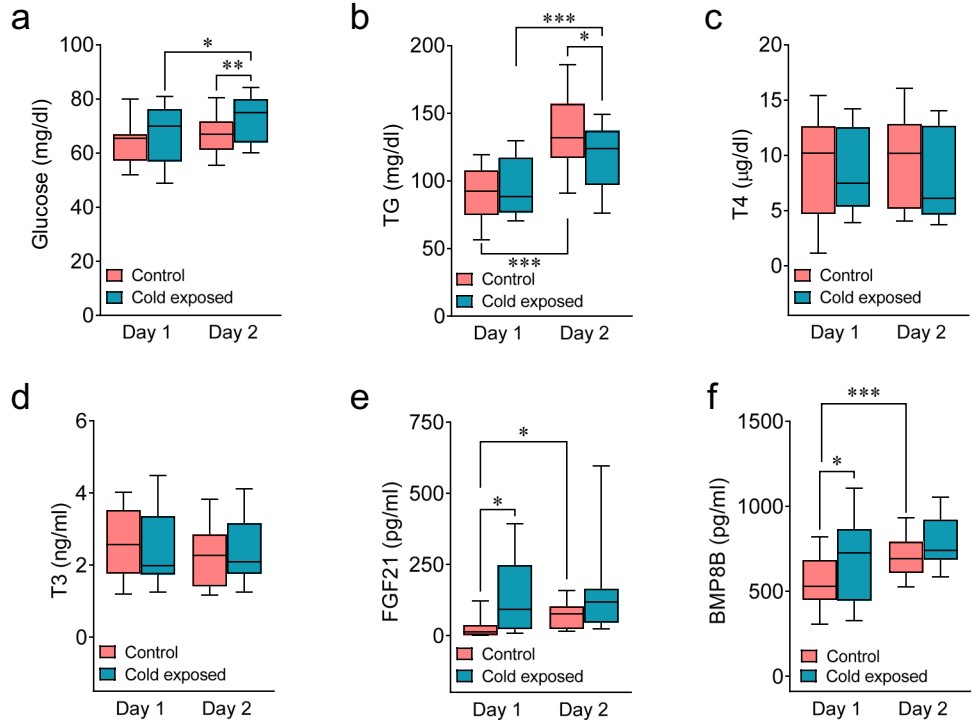

**Fig. 3 Circulating parameters in newborns after the cold stimulus. a–f** Circulating levels of glucose, TG, T4, T3, FGF21, and BMP8B in newborns at postnatal days of life 1 or 2, under control or cold exposed conditions. Number of newborns/group (order of groups): **a** 24, 26, 24, and 26, **b** 24, 22, 19, and 23, **c** 24, 26, 22, and 24, **d** 24, 26, 24, and 25, **e** 13, 18, 14, and 19, **f** 18, 15, 16, and 15. Box plots indicate median (middle line), 25th, 75th percentile (box), and 10–90th percentiles (whiskers; minima and maxima, respectively). Glucose cold exposed day 1 vs. glucose cold exposed day 2 $P = 0.007$; Glucose control day 2 vs. glucose cold exposed day 2 $P = 0.039$; TG control day 1 vs. TG control day 2, $P < 0.0001$; TG cold exposed day 1 vs. TG cold exposed day 2, $P < 0.0001$; TG control day 2 vs. TG cold exposed day 2 $P = 0.047$; FGF21 control day 1 vs. FGF21 cold exposed day 1 $P = 0.025$; FGF21 control day 1 vs. FGF21 control day 2 $P = 0.05$; BMP8B control day 1 vs. BMP8B cold exposed day 1 $P = 0.023$; BMP8B control day 1 vs. BMP8B control day 2 $P = 0.001$. Statistical significance was determined by two-sided linear mixed models followed by contrast of marginal means differences, with $P$ values adjusted by Holm's method. Results are shown for interaction term (Treatment:Day). *$P < 0.05$ or $P = 0.05$ (for FGF21) **$P < 0.01$, ***$P < 0.001$. Statistical parameters for panels **a–f** are summarized in Supplementary Table 3. Source data are provided as a Source data file.

following the second exposure (postnatal day 2) than the first one (postnatal day 1). Our data confirmed this hypothesis, demonstrating that these babies maintain a higher body temperature on day 2 than on day 1 after the cold stimulus (Fig. 7a–c). Notably, control babies did not show the adaptation to the cold stimulus (day 1: −0.36 ± 0.04 °C; day 2: −0.40 ± 0.06 °C; $P = 0.4112$, non-significant) (Supplementary Fig. 6a), suggesting that the response observed in cold exposed babies is likely dependent on the previous stimulus and not on age.

To gain more insight on the possible role of BAT thermogenesis and on the potential mechanism mediating this effect, we analyzed the correlations between changes in BAT temperature and the circulating factors at postnatal day 2. Whereas glucose, TG, T4, T3, and FGF21 did not exhibit any correlations with BAT temperature changes (Fig. 7d–h), BMP8B displayed a highly significant positive association with it (Fig. 7i). Taken together this evidence would suggest that (i) cold exposure at the very early stages of postnatal life promotes BAT thermogenic responses leading to a better defense of body temperature in subsequent cold events and ii) that this effect is associated with the circulating levels of BMP8B, but not with other well-known thermogenic hormonal signals acting in adulthood. Therefore, BMP8B is likely to play a role in the setup of thermogenesis in neonatal humans.

## Discussion

Here, we report a functional link between BAT thermogenesis and regulation of body temperature in human newborns. We also

demonstrate that BMP8B, but no other well-known thermogenic hormones, is associated with this process, suggesting a possible role in the development of the thermogenic programs in postpartum. This evidence is of importance to understand neonatal hypothermia, a critical condition in which physiopathological mechanisms remain largely unknown[1–4].

BAT is a specialized tissue responsible for heat production through non-shivering facultative thermogenesis[7–12]. Until recently, BAT was considered relevant only in small or hibernating mammals and in newborn humans[7,8,12,17–20]. However, regardless of the considerable amount of evidence on BAT thermogenesis and its functional relevance in adults, the current knowledge about brown fat in human newborns has not substantially evolved in the last 50 years[33–36]. In this study, we have used state-of-the-art IRT as a non-invasive tool for screening and functional analyses of BAT thermogenic activity in healthy neonates[25]. This method has been extensively used by us and others in the investigation of brown fat function in animal models and adult humans[21–26], but although its utilization in neonatal infants is less extended, it has been demonstrated to be effective for analyzing temperatures in different body parts at that age[25]. In this regard, although IRT is a good surrogate for the estimation of BAT temperature, layer subcutaneous fat insulation may vary in its thickness in adult humans, and therefore this could affect the IRT of BAT[37–42]. However, it must be taken into account that white fat depots in newborns are scarce and the skin is very thin[27]. Moreover, the advantages of the method: (i) non-invasive approach, (ii) sensitivity to cold simulation, (iii) lack of radiation

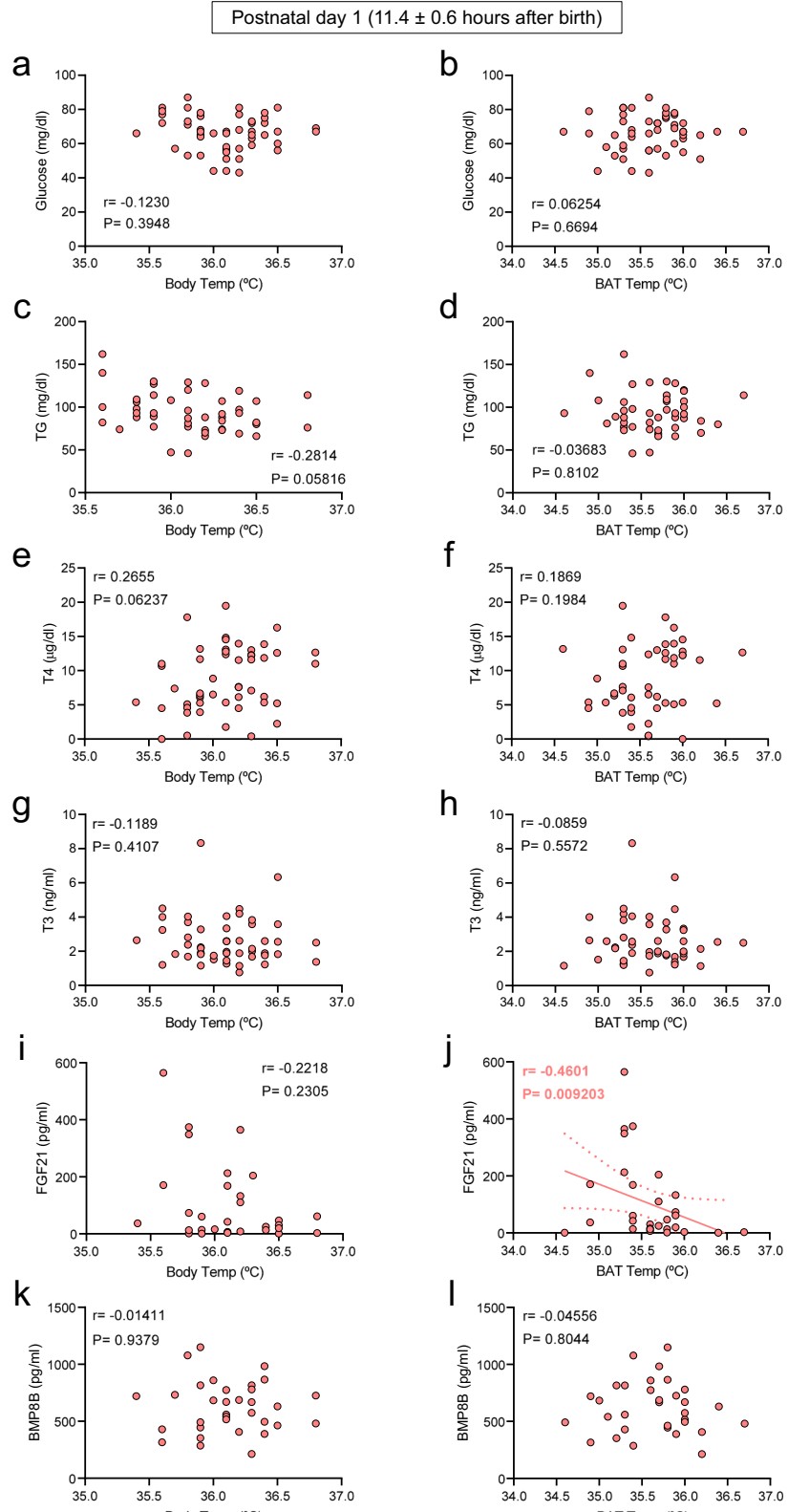

**Fig. 4 Correlations between body and BAT temperatures and circulating parameters in control 1-day old newborns.** Correlation between body temperature (**a**, **c**, **e**, **g**, **i**, **k**) and BAT temperature (**b**, **d**, **f**, **h**, **j**, **l**) in newborns in control conditions at postnatal day 1. Number of newborns: **a** 50, **b** 49, **c** 46, **d** 45, **e** 50, **f** 49, **g** 50, **h** 49, **i** 31, **j** 31, **k** 33, and **l** 32. Association analysis was performed by two-sided Pearson's test or Spearman's test; regression line with 95% confidence interval was added when correlation was significant. Source data are provided as a Source data file.

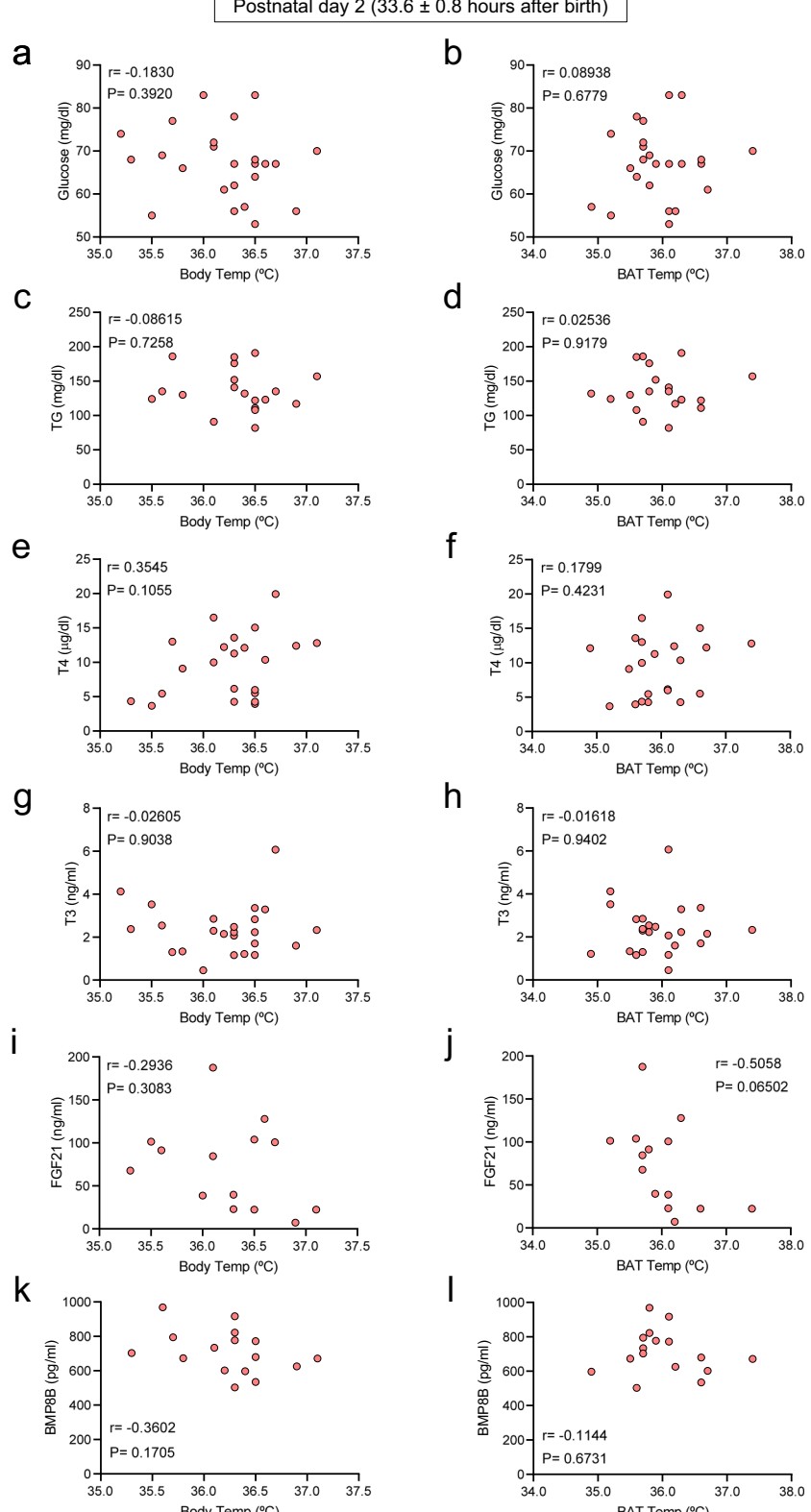

**Fig. 5 Correlations between body and BAT temperatures and circulating parameters in control 2-day old newborns.** Correlation between body temperature (**a**, **c**, **e**, **g**, **i**, **k**) and BAT temperature (**b**, **d**, **f**, **h**, **j**, **l**) in newborns in control conditions at postnatal day 2. Number of newborns: **a** 24, **b** 24, **c** 19, **d** 19, **e** 22, **f** 22, **g** 24, **h** 24, **i** 14, **j** 14, **k** 16, and **l** 16. Association analysis was performed by two-sided Pearson's or Spearman's tests. Source data are provided as a Source data file.

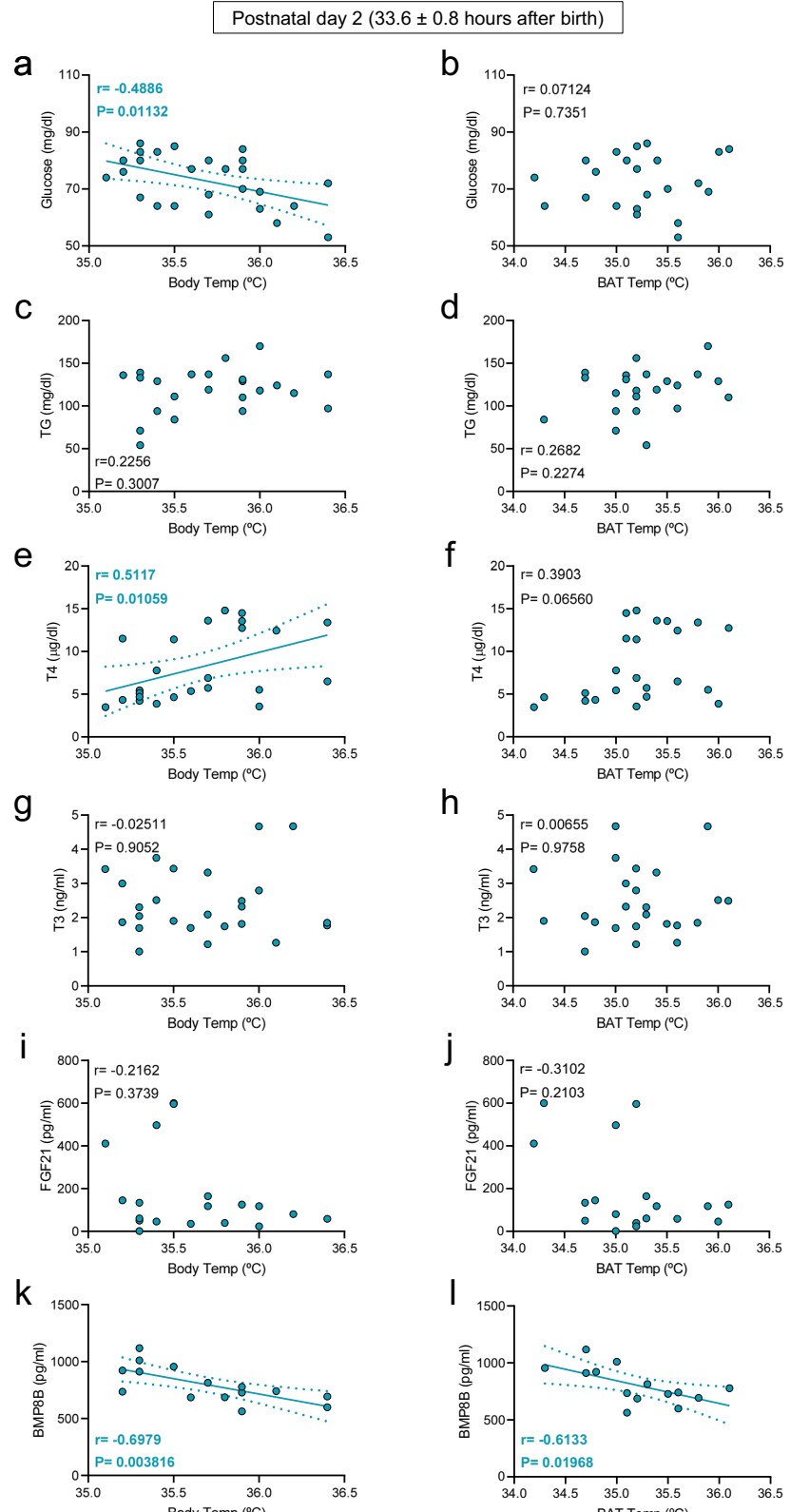

**Fig. 6 Correlations between body and BAT temperatures and circulating parameters in cold exposed 2-day old newborns.** Correlation between body temperature (**a**, **c**, **e**, **g**, **i**, **k**) and BAT temperature (**b**, **d**, **f**, **h**, **j**, **l**) in cold exposed newborns at postnatal day 2. Number of newborns: **a** 26, **b** 25, **c** 23, **d** 22, **e** 24, **f** 23, **g** 25, **h** 24, **i** 19, **j** 18, **k** 15, **l** 14. Association analysis was performed by two-sided Pearson's test or Spearman's test; regression line with 95% confidence interval was added when correlation was significant. Source data are provided as a Source data file.

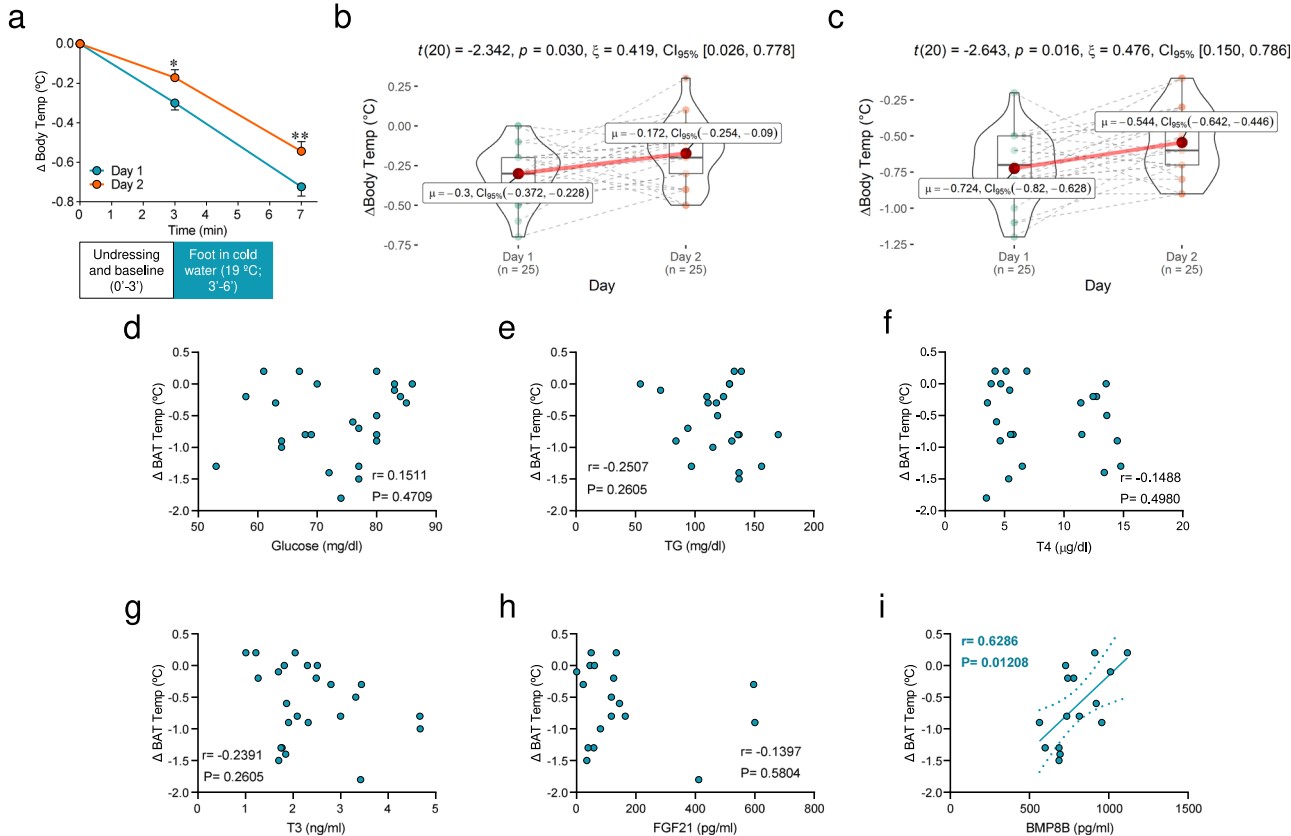

**Fig. 7 Cold exposure improves a subsequent thermogenic response in newborns. a** Body temperature changes on day 1 and day 2 in cold exposed newborns. A number of newborns/group: 25. Data represent mean ± SEM. $t = 3$ day 1 vs. $t = 3$ day 2 $P = 0.031$; $t = 7$ day 1 vs. $t = 7$ day 2 $P = 0.0061$. **b** Violin plot showing the individual body temperature changes between minute 3 and minute 7 at day 1 and day 2 in cold exposed newborns. A number of newborns/group: 25. Violin plots indicate median (middle line), 25th, 75th percentile (box) and $Q1 - 1.5\times$ IQR (interquartile range) and $Q3 + 1.5\times$ IQR (whiskers; minima and maxima, respectively). $P = 0.03$. **c** Violin plot showing the individual body temperature changes between minute 0 and minute 7 at day 1 and day 2 in cold exposed newborns. A number of newborns/group: 25. Violin plots indicate median (middle line), 25th, 75th percentile (box) and $Q1 - 1.5\times$ IQR (interquartile range) and $Q3 + 1.5\times$ IQR (whiskers; minima and maxima, respectively). $P = 0.016$. Statistical significance for panels **a–c** was determined by two-sided linear mixed models followed by contrast of marginal means differences, with $P$ values adjusted by Holm's method. $^*P < 0.05$, $^{**}P < 0.01$. **d, i** Correlation between BAT temperature change and circulating factors in cold exposed newborns at postnatal day 2. Number of newborns/group: **d** 25, **e** 22, **f** 23, **g** 24, **h** 18, **i** 15. Association analysis was performed by two-sided Pearson's test or Spearman's test; regression line with 95% confidence interval was added when correlation was significant. Source data are provided as a Source data file.

exposure, and (iv) control of variables (in this study: age, weight, and room temperature) make IRT likely the safest method for the investigation of BAT function in such a sensible and vulnerable human population, as newborn babies[40,41]. In this sense, IRT has been validated and compared with PET-CT using $^{18}$F-FDG and concluded that IRT is a convenient technique for studying BAT function[38,39,43].

Our data clearly demonstrate that the temperature of the interscapular area in human newborns ranges between approximately 34 and 38 °C and importantly, that it significantly correlates with body temperature in boys and girls. Moreover, this association does not reveal any gender dimorphism since the correlation is similar in both male and female neonates. To gain deeper functional insights on the thermogenic effects of neonatal BAT and considering the methodological and ethical constraints that imply the experimental work with healthy newborns, we applied a short and mild thermogenic stimulus. Despite the short-term timing (3 min) of this cold stimulus, it was enough to provoke relevant changes. Firstly, it significantly decreased peripheral body temperature in an average of almost 0.5 °C (reaching −1.0 to −1.2 °C in some newborns). Secondly, despite that marked outcome on body temperature, the temperature reduction average of the interscapular BAT area decreased was less

than 0.1 °C, indicating that thermogenic activation defended it against a bigger drop. Thirdly, and more importantly, the early exposure to cold (postnatal day 1) led to a better defense of body temperature in subsequent cold events (postnatal day 2). Together, these results support the idea that BAT thermogenesis regulates body temperature and cold adaptation in neonatal infants. A possible limitation of our study could be the fact that the temperature of the neonatal room was lower than the thermoneutral zone for newborns (32–35 °C). However, it is important to maintain a balance between an optimal thermic environment and hyperthermia[3,4,25,44–49]. In our Hospital, as in many others, the temperature, for clinical, scientific, and ethical reasons, was set at 26–28 °C in the delivery room and at 25–26 °C in the maternity area, both within the range suggested by WHO[47,48]. Therefore, despite being outside of the thermoneutral range, the environmental temperature was constant and affected both control and cold exposed newborns, limiting its potential stressful effects.

Several hormones modulate BAT activity, by directly exerting their effects on brown adipocytes and indirectly, acting on the hypothalamus to control the firing of sympathetic nerves[7,8]. Among them, THs (T3 and T4)[28,50], FGF21[51,52], and BMP8B[21,53] playing an important role, in both rodent models and/or adult

humans. At postnatal days 1 and 2, the control children did not show any significant association with any of the hormonal parameters assayed, glucose or TG. However, on day 2 we found that body temperature had a negative correlation with glycemia in the cold exposed babies. This data may indicate that the maintenance of higher temperature would require a greater glucose demand by the BAT, suggesting its possible implication as to the primary fuel for thermogenesis at very early neonatal stages, in keeping with the scarce amount of WAT at this stage of life[27]. We also found that both body and BAT temperatures positively correlated or trend to correlate with the circulating levels of T4. It has been proposed that the acute surge in THs that occurs at birth may have limited significance in neonatal thermogenesis[54], however, our data would go in the opposite direction, suggesting that T4 production could be associated with thermogenesis in newborns, although not as much as BMP8B.

BMPs belong to the transforming growth factor β superfamily (TGF-β). BMPs regulate a wide range of processes from embryonic development to tissue homeostasis[55,56]. Recent data have revealed a thermogenic role for BMP8B, which has been demonstrated to be produced in the brown fat (therefore, being considered as a batokine) and to induce thermogenesis by acting both peripherally and centrally on the hypothalamus[21,53]. BMP8B involvement in thermogenesis in humans and more precisely in newborns remained unclear, however, the evidence presented in this study supports this notion; this hypothesis being demonstrated by three facts. Firstly, postnatal time and cold exposure increased the circulating levels of BMP8B in neonates, suggesting its involvement in the initial stages of thermogenesis. Secondly, a highly significant and negative correlation between body and BAT temperatures with circulating BMP8B levels was demonstrated: those babies with lower temperature display higher BMP8B levels, indicating the effect of cold induction. Lastly and more importantly, BMP8B correlates with the changes in BAT temperature stimulated upon cold induction, suggesting its role in human neonatal thermoregulation.

In summary, this study demonstrates that BAT thermogenesis correlates with body temperature in newborns and that cold exposure is a key event activating functional and endocrine responses modulating brown fat adaptation. Our data also point to BMP8B associated with this process. Overall, this work will help to understand the setup of the thermogenic mechanism in humans, which is of interest to avoid neonatal hypothermia and its life-threatening consequences.

## Methods

**Subjects**. A cohort of 50 white newborns was used for this study (Table 1). All these babies were recruited at the Neonatology Service of the University of Santiago de Compostela Hospital Complex (CHUS). The parents or legal representatives of all newborns gave their written informed consent, which was validated and approved by the Ethical Committee and the Committee for Clinical Investigation of the CHUS (code 2015/079). Thermal images of newborns are published with the informed consent of their parents or legal representatives. The study design and conduct complied with all relevant regulations regarding the use of human study participants and was conducted in accordance with the criteria set by the Declaration of Helsinki.

**Cold exposure protocol**. Two experimental groups were established (Fig. 2A): (i) cold exposed newborns to whom a thermogenic stimulus was applied and (ii) controls who did not undergo any thermogenic stimulation. The thermogenic stimulus consisted of the immersion of the right foot in 19 °C (18–20 °C interval) water for 3 m. This protocol was designed to reach a temperature decrease between a mild and upper range of moderate hypothermia according to WHO criteria[47,48]. The tests of all the cases were always carried out in the same room, in which the temperature was maintained at 25–26 °C. In both groups, the test began by undressing the newborn, measuring both peripheral body and BAT temperature, and leaving him/her only with the diaper for 3 min; the objective of this was to homogenize the basal conditions for all the babies because not all were wrapped in the same way. Once the thermogenic stimulus was applied, the babies remained undressed for 5 min, after which the protocol was considered completed. During

this time, the peripheral body temperature and the temperature of the BAT were measured at minute 0 (when undressed), 3 (when applying the thermogenic stimulus), 7 (1 min after thermogenic stimulus ended because this was the moment when the babies reached the lowest body temperature), and 11 (when the protocol finished). In the control group (absence of thermogenic stimulation), the protocol finished at minute 3. At the end of the setting, in both groups, a blood sample was obtained for the determination of glucose, TG, total T4, total T3, TSH, FGF21, and BMP8B. The test was performed twice: at day 1 of life (11.4 ± 0.6 h after birth) and at day 2 of life (33.6 ± 0.8 h after birth) in the same individuals. One of the objectives of this study was to compare the baseline values of these determinations with the values obtained after the activation of the BAT. Given that performing two blood extractions on the same newborn 3 min apart (before and after the thermogenic stimulus) without a justifiable clinical reason, it was not considered ethically plausible, therefore, the blood samples obtained at minute 3 were taken as baseline values. None of these babies had blood extraction exclusively for this study. All of them were healthy newborns to whom, following the protocol of our center, underwent routine, serial tests at 6–12 h of life and 24 h later. Infants with temperature (i.e., fever) or analytical abnormalities were not included in the temperature analyses.

**Temperature measurements**. A digital thermometer (ThermoTester 082.030A; Laserliner; Arnsberg, Germany) with a scale of 0.1 °C was used to measure the water temperature. Peripheral body temperature was measured in the posterolateral (deltoid) area of the right arm with an infrared contact thermometer (Intersurgical; Madrid, Spain) with a resolution of 0.1 °C scales and accuracy of ±0.2 °C. BAT temperature was recorded in the interscapular area with an infrared camera (E60bx: Compact-Infrared-Thermal-Imaging-Camera; FLIR; West Malling, Kent, UK). In relation to the emissivity ($\varepsilon$), the values that have been reported for the emissivity of the human skin range from 0.91 to 0.99, with a consensus of $\varepsilon = 0.98 \pm 0.01$[43,57–62], when mentioned, as not many reports specify this parameter in the current literature. However, this consensus value, in our view had a potential weakness in the case of the subjects of our study, namely human newborns. The value of $\varepsilon = 0.98$ is based on a multitude of studies that have analyzed emissivity in adult humans, mainly male and body areas, such as fingers, back and palm of the hand, wrist, forearm, toes, and forehead[43,57–61]. Notably, those areas with thicker skin, such as hands, outer wrist, dorsal forearm, and feet show higher emissivity than those other areas with thinner skin, such as cheek, inner wrist, and volar forearms[43,57–61,63]. Moreover, recent data have also shown that women, having slimmer skin compared to men, have a lower emissivity (0.046 as an average) than men[58]. For these reasons, given the thin skin of human newborns[27,64] and despite that the $\varepsilon = 0.98$ has been used before in other neonatal studies[40,41,65,66], we considered a lower value of $\varepsilon = 0.95$, which is still in the range of human skin emissivity. In any case, we extensively tried different values of emissivity when optimizing our protocols. The analyses were also done with a value of $\varepsilon = 0.98$; the obtained data/correlations were similar, and the conclusions of the study were the same.

Babies were placed in a prone position on a fat mattress, as shown in Figs. 1a and 2b. They were not held by anyone. Focus distance was set at the interval 40–50 cm. The interscapular region was defined between the scapulae or shoulder blades, an area where BAT is abundant in newborns[11–13]. In this region, a mass of BAT is found under the subcutaneous WAT[11–13], and therefore the temperature of the skin overlying it provides a reliable and accurate readout of BAT temperature and activation. Besides BAT, thermographic analyses were also performed on (i) the neck, more precisely on the upper part of the nuchal fold which was selected to avoid artifacts related to the skin bending, and (ii) on the outer part of the ear, more precisely in the scapha. The analysis of the images was performed with the FLIR Tools Software Package (FLIR; West Malling, Kent, UK)[21,24] and in all the cases average temperature of the selected area was chosen. The size of that area and the landmarks were similar for all the babies. BAT temperature was not adjusted by weight, length of BMI, because as noted in Table 1, the variations in those parameters were minimal (3290.2 ± 51.0 g, 49.7 ± 0.3 cm, and 13.3 ± 0.1 kg m$^{-2}$, respectively). Neither body nor BAT temperature correlated with bodyweight or BMI (Supplementary Figs. 4). Temperature (BAT and other body parts) was expressed in °C.

**Blood extraction**. All the newborns in this study were sampled. Blood was extracted by venepuncture, which is the best method for extracting blood from neonates, as recommended by WHO[67], from the dorsal part of the hand using a winged steel needle (23 gauge), with an extension tube (a butterfly). The volume of collected blood was around 6 ml for each subject (3 ml for day 1 + 3 ml for day 2), corresponding to approximately 3% of the total blood volume, within the limits specified by WHO[27,67]. From these 3 ml of blood sample (for each time point), 1.5 ml were used for analytic routine analyses in the Neonatal Service. The remaining 1.5 ml were used for this study. Considering the hematocrit of a newborn infant (55%)[27], approximately, 675 μl of serum per baby per time point were obtained.

**Blood biochemistry**. For the analytical determinations, blood samples were collected (1.5 ml, EDTA-containing tube) and subsequently centrifuged (15 min, 1500$g$) to

isolate the serum. The samples were aliquoted and frozen at −80 °C. Glucose was measured with a glucometer (Accu-check; Roche; Barcelona, Spain) and TG was analyzed using the GPO-Trinder Method in the ADVIA 2400 Chemistry System (Siemens; Erlangen Germany). Plasma hormonal parameters were analyzed using the commercial kits: T4 (EIA-1781; sensitivity: 0.5 μg dl$^{-1}$; intra-assay coefficient of variation (CV): 3.1-4.3%; interassay CV: 2.4–4.5%; DRG Diagnosis; Marburg, Germany), T3 (EIA-1780; sensitivity: 0.2 ng ml$^{-1}$; intra-assay CV: 3.2–9.6%; interassay CV: 1.2–10.3%; DRG Diagnosis; Marburg, Germany); TSH (AutoDELFIA® Neonatal hTSH, B032-312; sensitivity: 2 μU ml$^{-1}$; intra-assay CV: <6%; interassay CV: 1.2–1.5%; Perkin Elmer; Boston, MA, US); FGF21 (HLPPMAG-57 K; sensitivity: 0.003 ng ml$^{-1}$; intra-assay CV: <10%; interassay CV: <10%; EMD Millipore Corporation, Billerica, MA, USA) and BMP8B (MBS944757; sensitivity: 2.34 pg ml$^{-1}$; intra-assay CV: <8%; interassay CV: <10%; MyBiosource; San Diego, CA, USA). Samples were analyzed simultaneously in replicates for each individual.

**Statistical analysis**. Data are expressed as mean ± SEM and analyzed with the statistical software GraphPad Prism 8.0.2. and packages lme4[68] and emmeans[69] from free software R[70]. Statistical significance (two-sided in all the cases) was determined by Mann–Whitney test (when two groups were compared) or linear mixed model including an interaction term (when more than two groups were compared), correcting $P$ values with Holm's method. The relation between continuous variables was analyzed by simple correlation (Pearson's test or Spearman's test). The alpha value was set at 0.05.

**Reporting summary**. Further information on research design is available in the Nature Research Reporting Summary linked to this article.

## Data availability

The data that support the findings of this study are available from the corresponding author upon reasonable request. The source data underlying all Figures, all Supplementary Figures, all Tables, and all Supplementary Tables are provided as a Source data file. Source data are provided with this paper.

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

## Acknowledgements

Miguel López thanks María López-Piñeiro, Sara López-Piñeiro, Marta López-Piñeiro, and Irene Piñeiro for the opportunity to do this work and because he would never have done this study without them. He also thanks all the clinical staff of Neonatology Service, of University Clinical Hospital of Santiago de Compostela for their outstanding support, help, and assistance during November 2014. The research leading to these results has received funding from the Xunta de Galicia (R.N.: 2016-PG057; M.L.: 2016-PG068); Ministerio de Economía y Competitividad (MINECO) co-funded by the FEDER Program of EU (R.N.: RTI2018-099413-B-I00; C.D.: BFU2017-87721-P; F.V.: SAF2017-85722-R; M.L.: RTI2018-101840-B-I00: F.V., J.M.F.-R., and M.L.: BFU2017-90578-REDT/Adipoplast); Instituto de Salud Carlos III (J.M.F.-R.: PI15-01934); Atresmedia Corporación (R.N. and M.L.); Fundación BBVA (R.N.); la Caixa Foundation (ID 100010434), under the agreement LCF/PR/HR19/52160022 (M.L.); European Foundation for the Study of Diabetes (R.N.) and ERC Synergy Grant-2019-WATCH- 810331 (R.N.) and Research Grant SOPEGA 2016 (A.U. and M.L.C.). F.V. is an ICREA Academia researcher (Generalitat de Catalunya). I.G.-G. is the recipient of a fellowship from the European Union's Horizon 2020 research and innovation program under the Marie Sklodowska-Curie actions. The CiMUS is supported by the Xunta de Galicia (2016-2019, ED431G/05). CIBER de Fisiopatología de la Obesidad y Nutrición is an initiative of ISCIII. The funders had no role in study design, data collection, and analysis, decision to publish, or preparation of the paper.

## Author contributions

A.U., N.L., and N.M. performed the neonatal protocols, collected the temperature data and the blood samples. I.G.-G. and A.E.-S. performed the blood biochemistry. I.G.-G., M.P.P., and M.L. analyzed the data. M.P.P. and M.L. performed the statistical analyses. A.U., I.G.-G., M.L.C., and M.L. designed the experiments. A.U., I.G.-G., E.M., C.G.-M., C.A.S., R.N., C.D., F.V., J.M.F.-R., M.L.C., and M.L. discussed and interpreted the data. M.L. is the senior author and lead contact of this work, he was the first experimental observations, developed the hypothesis, secured funding, supervised and directed the project, made the figures, and wrote the paper. All authors reviewed and edited the manuscript and had final approval of the submitted paper.

## Competing interests

The authors declare no competing interests.
