## [Peer Review File · Nature Communications]

BMP8 and activated brown adipose tissue in human newbornsReviewer #1 (Remarks to the Author):

This is a limited study as not only is there only the briefest of descriptions of the thermal imaging and subsequent data analysis, but from the small number of example images included it is clear that the distance from the camera to the subject varies. This is a critical aspect of conducting robust thermal imaging studies, as any variation in distance from the subject and camera will result in differences in temperature and thus be the most probable explanation for the correlations reported.

Reviewer #2 (Remarks to the Author):

In this manuscript, Dr. Miguel Lopez and colleagues used non-invasive infrared thermography (IRT) to investigate human neonatal brown adipose tissue (BAT) activity. This group and others had previously reported that IRT is a suitable non-invasive, safe, and quick method to estimate BAT thermogenic activation, also in humans (see e.g., references 23-25 in the manuscript; DOI: 10.1007/s12576-016-0472-1; DOI: 10.1371/journal.pone.0220574; DOI: 10.1002/oby.22560). However, until now, data on BAT thermogenesis in newborns are lacking. For the first time, the authors show that mild cold stimulus promotes BAT activation in healthy newborns, and that one previous single short-term cold stimulus in the first postnatal day improves body temperature adaptation in a subsequent cold event at day 2 of postnatal life. Moreover, the authors analyzed circulating levels of several metabolites and hormones known to be involved in BAT activity regulation (thyroid hormones, FGF21 and BMP8B). Both postnatal time and cold exposure increase the circulating levels of FGF21 and BMP8B in neonates. Moreover, the batokine BMP8B correlates with the changes in BAT temperature stimulated upon cold induction. Thus, the present study reveals an endocrine role for the neonatal human BAT, which will be of great interest to the researchers in the adipokine field. In fact, the recently recognized secretory role of BAT and its involvement in whole-body crosstalk, also in humans, is an emerging issue here and now (ref 11 in the manuscript; DOI: 10.1016/j.tem.2017.10.005; DOI: 10.1038/nrendo.2016.136). However, current reports on batokines are mostly based on studies in mouse models, while data on human batokines are limited. Overall, the uniqueness of the data presented in the manuscript is highly relevant and results are discussed appropriately in relation to the currently available literature. The reviewer has a few suggestions that could improve the quality of data presentation and interpretation if implemented.

-IRT measurements in neonates are critical results in the manuscript. However, given the level of detail provided in the Methods section, the ability of any researcher to reproduce these measurements would be low. Neonates underwent infrared thermography to determine the temperature of the skin overlying the main superficial BAT depot in the interscapular region. However, how this anatomical region was defined for the measurements is not detailed. Nor is it specified which parameter was measured as BAT temperature: the maximum temperature in the defined area or the mean temperature in the area? Although the infrared camera equipment, the software used for image analysis, and corresponding references are well stated in the manuscript, a carefully detailed protocol for IRT measurements must be provided.

-In general, statistical analyses and level of detail provided in Methods are adequate. However, the variability in the number of individual data shown in the correlation analysis graph for each of the hormonal parameters should be clarified.

-Figure 2. (A) The diagram for the cold exposure protocol is very useful. However, the control diagram, also depicting an 11-minute bar, is somewhat misleading. (C) It should be defined how were calculated the body and BAT temperature changes at the cold exposed condition since different measures were determined (at minute 7 or at minute 11). (D-I) Data are presented in histograms. In the legend it should be better indicated as 'Data are means +/- SEM' and not 'Center values represent average; error bars represent SEM' as stated. Also check the definition of the statistical significance comparison for each of the symbols.

-Figure 6. Which is the difference between data in (B) and (C) (i.e., violin plots showing the individual body temperature changes at day 1 and day 2, as stated in the corresponding figure legend)? (D-I) Correlation analyses between BAT temperature change and circulating factors in cold exposed newborns at postnatal day 2. It should be indicated how BAT temperature change was calculated.

-Headings of the figure legends should be changed since no direct 'effects' were studied.

- There is a repeated reference (the reference in numbers 19 and 28 is the same).
- Reference 11 is not complete.

Reviewer #3 (Remarks to the Author):

The manuscript „BMP8B sets up brown fat thermogenesis in newborns“ by Urisarri et al. studies early postnatal brown fat thermogenesis in humans, aiming to unravel the underlying molecular mechanisms and the possible benefits. The rationale of the study is of great importance and timely given the vast interest in brown fat at the moment. Most importantly, human data on the topic are scarce and urgently needed. The manuscript is well written, the design of the study very carefully done considering the ethical issues working with human newborns, and the data are original and of high relevance. However, there are some technical issues that need to be addressed.

1) The major problem with peripheral cold stimulation (also in adult humans) is the fact that in addition to potentially triggering BAT, it also induces a massive vasoconstriction in the extremities (as seen in Fig 2C). This lead to a shunting of the circulation and thereby potentially a maintenance of core body temperature, which could be misinterpreted as BAT thermogenesis. In the current study, body temperature was only recorded by contact infrared thermometer under the arm, which likely provides values that drop below body temperature upon cold stimulation due to reduced blood flow. Therefore a second more independent measurement would be required, ideally rectal temperature. I understand that this might be difficult in newborns also from an ethical perspective, but the authors have infrared pictures available showing the ear temperature. While this might not be true inner ear temperature, it would probably still work as a proxy of body temperature that is less affected by vasoconstriction. Also the neck region, where the skin is very thin, might be suitable. I would therefore recommend reading also the temperature data from these areas and then calculate a BAT/ear temperature value for each individual to assess BAT activation.

2) I am not an expert in statistics, but it seems to me that the results in Fig 2d-i have a design with 2 independent factors (cold, time), so a 2-way repeated measure ANOVA would be appropriate for the analysis, which would also provide an interaction p-value that could be of interest.

3) It is unclear how the BAT temperature was calculated. Was it the mean temperature of the selected area or the maximum value of the area? If it was the first case, was the area somehow adjusted to newborn weight or size?

4) The discussion reads a bit unfocused and extensive. It should be shortened to maintain the focus on the data provided.

5) The conclusions on the molecular mechanisms depend heavily on the quality of the ELISA used in the study. The authors should provide information on the inter- and intraassay variance of these kits (were all samples measured in one run?), as well as publications where these kits were validated. Did the authors also check TSH levels as this might be a more sensitive readout of thyroid function than total T4 and T3? It should also be added that the kits measure total hormone and not free hormone.

EDITOR

Response: We want to thank the Editor and the Reviewers for the helpful and encouraging comments and insights, which have been instrumental to produce a substantially improved version of our manuscript. Secondly, we would like to apologize for the slight delay in the resubmission of the revised manuscript. In this new version, we have addressed all the points raised and, besides the discussion requested, we have included, following the advice of the Editor and Reviewers, new analyses to reinforce our original data. We sincerely believe that the new results substantially strengthen the idea that BAT temperature correlates with body temperature and that mild cold stimulus promotes BAT activation in newborns. Overall, our data uncover key features of the setup of BAT thermogenesis in newborns. A detailed point-by-point response to all comments of the Referees is included below. Please, note that throughout the rebuttal the **Figure numbers** refer to the new version of the manuscript. Please, note also that former **Suppl. Table 2** has been removed because all that information has been included in the submitted **Source Data** Excel file.

Summary of the changes

In addition to clarifying the questions raised, we provide **20 new experimental set of data:** analysis of ear and neck temperature (and correlations with BAT temperature), analysis of BAT temperature at different distances, ratios of BAT temperature, correlations between temperature with feeding (breastfeeding or formula) and TSH circulating levels (and correlations with BAT temperature). These new analyses result in the **Figures 1e-j, Figure 2d-g, Suppl. Figure 1a-e Suppl. Figure 2a-e and Suppl. Table 1** to answer the concerns of the Reviewers and to support our claims. We strongly believe that the new evidence, briefly summarized below, substantially strengthen the manuscript.

1. In response to **Reviewer#1**, we would like to state that we always control the distance from which we take the pictures of the babies; this was set to 45 ± 5 cm. Therefore, there is not source of variability here. In any case, we have performed a double strategy:
 - a) Analysis of BAT temperature by thermography at different distances (10, 25, 50 and 75 cm; **Suppl. Figures 1a-e**).
 - b) Theoretical explanation of this effect exposed in pages 2-3 of this rebuttal.
 2. In response to **Reviewer#2**, we have clarified the questions raised:
 - a) Explanation of the infrared thermography IRT measurements (i.e. **Figure 2**), software details.
 - b) Clarification of the requested statistical details (i.e. **Figure 6**).
 - c) Improvement of the cold exposure protocol diagram (**Figure 2a**).
 - d) Formal issues (headings and references).
 3. In response to **Reviewer#3**, we have clarified the questions raised and performed additional analyses/changes:
 - a) Temperature in other body areas (neck and ear, as requested) and their correlation with BAT temperature (**Figures 1e-j and Figures 2d-g**).
 - b) Statistical improvement of the analysis of circulating parameters (**Suppl. Table 2**).
 - c) Explanation of the IRT measurements (i.e. **Figure 2**), software details.
 - d) Better focus of the discussion, which has been shortened.
 - e) Measurement of the TSH values and correlation with BAT temperature (**Suppl. Figure 2a-e**)
-

REVIEWER#1

General comment: *This is a limited study as not only is there only the briefest of descriptions of the thermal imaging and subsequent data analysis, but from the small number of example images included it is clear that the distance from the camera to the subject varies. This is a critical aspect of conducting robust thermal imaging studies, as any variation in distance from the subject and camera will result in differences in temperature and thus be the most probable explanation for the correlations reported.*

Response: We thank the Reviewer#1 for his/her insight on our manuscript. This is, actually, an excellent question. For the current work, we have taken and analyzed almost 2,000 thermographic images from newborns. One of the first parameters that we fixed was the distance of focus. For practical reasons (height of the staff, position of the babies, facilities where the images were taken etc.), **the focus distance was set at 40-50 cm and we kept this factor very rigorously**, since, as Reviewers states, we were also afraid that it could be a confounding factor. The 10 cm variation (40-50 cm) was basically due to the difference in height of the 3 different people taking the pictures. We deeply apologize for not including this information in the former version of the manuscript; it has been included in the new version. The only two images that were taken at smaller distance are those shown in the **former Figure 1a**, with a focus distance of 25 cm (please, compare with **Suppl. Figure 1a** of the new version), because we wanted to highlight the interscapular area where the BAT (and in this new version neck an ear area, as requested by **Reviewer#3**), measurements were taken. To avoid misperception, in the new version we have replaced those two images for a new one (**Figure 1a**).

In any case, despite all the above considerations, in light of your comment, we decided to further address this issue from both the theoretical and the experimental point of view. As explained now in detail, we provide clear evidence that in the range of distances used in our experiments changes in the **distance does not interfere in the thermographic acquisition, recording and analysis of the data**. This is proven by:

1. The new set of data included in the revised version (**Suppl. Figures 1a-c**). Those graphs show the temperature of the BAT area after taking pictures at different distances (10, 25, 50 and 75 cm) and demonstrate that the distance does not affect the BAT temperature area:
 - a) **Suppl. Figure 1a** shows 4 representative thermographic images of the same baby taken consecutively at 10, 25, 50 and 75 cm
 - b) **Suppl. Figure 1b** shows the average BAT temperature of several newborns. The average variation of a BAT temperature for the two extreme distances (10 cm and 75 cm) is 0.2 ± 0.1 °C in a 65 cm range, which can be considered negligible, in fact it is statistically non-significant.
 - c) **Suppl. Figure 1c** shows the BAT temperature of every newborn at the different distances. As shown in the graph, the BAT temperature variation is absolutely minimal along the focus distance range. Overall, these data demonstrate that the distance does not impact the thermographic recordings.
2. Providing further theoretical basis to our findings. Since addressing this issue in depth was well beyond our expertise as biologist or medics, we have collaborated and discussed our data and **Reviewer#1's** interesting concerns with **Prof. Carlos A. Salgado**, Director of the Institute of High Energy Physics (USC; <https://igfae.usc.es/igfae/es/>), also affiliated to CERN (Geneve, Switzerland; <https://home.cern/>). Prof. Salgado is a worldwide expert in Particle Physics and Quantum Physics (<https://scholar.google.it/citations?user=lOfaN2gAAAAJ&hl=it>), he is holder of an ERC Advanced Grant, as well as awardee of some of the most prestigious prizes in this field . He explained us that the radiation emitted by a given body does not depend either on the

distance to the observed or the detector. The heat that a given body emits (in this case a baby) is an infrared radiation and does not depend on the gap to the observer. In a simplistic way, the color that the thermographic camera records is always the same: the length of wave of the radiation cannot change from red (warm) to blue (cold) because of the distance of the observer. The theoretical basis of this principle is as follows:

The frequency of the electromagnetic energy radiated per unit volume for a body at temperature T is distributed according to the well-known **Planck's radiation law**:

$$\rho_T(\nu) d\nu = \frac{8\pi\nu^2}{c^3} \frac{h\nu}{e^{h\nu/kT} - 1} d\nu$$

proposed by Max Planck in 1900 and considered to mark the starting point of Quantum Physics. This equation remarkably shows that the energy density distribution of **radiation for a body of temperature T depends only on its temperature** and a set of universal constants, c the speed of light, k the Boltzmann constant, and h the Planck constant. This **radiation is independent on the material under consideration, the distance at which it is observed or any other quantity**. It is trivial to show that this distribution has a peak at the frequency

$$\nu_{\text{peak}} = \frac{3 + W(-3/e^3)}{h/k} \simeq 2.82 \frac{kT}{h}$$

where the last equality is obtained by solving W(z) as the solution of $z = w e^w$

This relation is known as the **Wien's law** and was derived in 1893, before the original work by Planck. Interestingly, it establishes a linear correspondence between the frequency of the electromagnetic radiation and the temperature of the body.

In other words, the electromagnetic radiation of a body (or the “color” of the light it emits) depends only on its temperature, the higher the temperature the larger the frequency of the radiation — or equivalently the energy of the radiated photons though the **Einstein relation**

$$E_\gamma = h\nu$$

Plugging the constants into the Wien's equation yields that the peak of the radiation for a typical temperature of 20-30 °C corresponds to wavelengths

$$\lambda_{\text{peak}} \sim 9.5 - 10\mu\text{m}$$

This wavelength sits in the infrared. The thermal imaging camera used for this study (*E60bx: Compact-Infrared-Thermal-Imaging-Camera; FLIR; West Malling, Kent, UK; <https://www.flir.com/about/about-flir/>*) is quoted to have a spectral range of 7.5 to 13 μm, optimized to measure temperatures in the range -20 to 120 °C. According to suppliers, inaccuracies in the determination of the temperature could arise when the image of the object forms only on few pixels of the detector - this may happen when the object is very, very far (well beyond the possible range variation in our experimental set up) from the camera. We have estimated that the interscapular area where the BAT temperature has been measured fills from 10% to 12% of the total size of the image, or 7600 to 9200 pixels. So, inaccuracies due to distant measurements can be ruled out. In summary, we deeply apologize for not being accurate in the first version and explaining that **pictures were always taken inside a set distance (40-50 cm)**.

In any case we thank the Reviewer for raising the subject, it is clearly a convenient and very interesting question. To know and understand to some extent the basis of why **radiation does not depend on distance, depends only on temperature** based on Quantum Physics it was a learning exercise very much appreciated by all the authors. In fact, if the editor agrees we will support including the theoretical argument in the Methods Section.

We thank again the **Reviewer#1** for his/her excellent insight and comments that have improved the scope and quality of our manuscript.

REVIEWER#2

Overall comment: *In this manuscript, Dr. Miguel López and colleagues used non-invasive infrared thermography (IRT) to investigate human neonatal brown adipose tissue (BAT) activity. This group and others had previously reported that IRT is a suitable non-invasive, safe, and quick method to estimate BAT thermogenic activation, also in humans (see e.g., references 23-25 in the manuscript; DOI: 10.1007/s12576-016-0472-1; DOI: 10.1371/journal.pone.0220574; DOI: 10.1002/oby.22560). However, until now, data on BAT thermogenesis in newborns are lacking. For the first time, the authors show that mild cold stimulus promotes BAT activation in healthy newborns, and that one previous single short-term cold stimulus in the first postnatal day improves body temperature adaptation in a subsequent cold event at day 2 of postnatal life. Moreover, the authors analyzed circulating levels of several metabolites and hormones known to be involved in BAT activity regulation (thyroid hormones, FGF21 and BMP8B). Both postnatal time and cold exposure increase the circulating levels of FGF21 and BMP8B in neonates. Moreover, the batokine BMP8B correlates with the changes in BAT temperature stimulated upon cold induction. Thus, the present study reveals an endocrine role for the neonatal human BAT, which will be of great interest to the researchers in the adipokine field. In fact, the recently recognized secretory role of BAT and its involvement in whole-body crosstalk, also in humans, is an emerging issue here and now (ref 11 in the manuscript; DOI: 10.1016/j.tem.2017.10.005; DOI: 10.1038/nrendo.2016.136). However, current reports on batokines are mostly based on studies in mouse models, while data on human batokines are limited. Overall, the uniqueness of the data presented in the manuscript is highly relevant and results are discussed appropriately in relation to the currently available literature. The reviewer has a few suggestions that could improve the quality of data presentation and interpretation if implemented.*

Response: We thank the Reviewer for the positive view of our manuscript and the excellent comments and suggestions. We also believe that the current manuscript provides novel and important data that enhances our understanding of the role of BAT in the regulation of temperature in neonatal stages. A detailed point-by-point response to the comments is included below.

Reviewer's comment 1: *IRT measurements in neonates are critical results in the manuscript. However, given the level of detail provided in the Methods section, the ability of any researcher to reproduce these measurements would be low. Neonates underwent infrared thermography to determine the temperature of the skin overlying the main superficial BAT depot in the interscapular region. However, how this anatomical region was defined for the measurements is not detailed. Nor is it specified which parameter was measured as BAT temperature: the maximum temperature in the defined area or the mean temperature in the area? Although the infrared camera equipment, the software used for image analysis, and corresponding references are well stated in the manuscript, a carefully detailed protocol for IRT measurements must be provided.*

Response: The Reviewer brings a very interesting point and a weakness of the former version that needed clear improvement. In the new version of the manuscript we have better explained the IRT protocol and the anatomical regions analyzed, as follows. The interscapular region was defined as the one between the scapulae or shoulder blades, an area where BAT is abundant in newborns^{1, 2, 3}.

This anatomical area is well-defined in the magnetic resonance image at the left (**Figure 3** from ²), where sagittal (**a**), coronal (**b**) and axial (**c**) planes are shown and the interscapular BAT is highlighted in green (scale bar 5 cm). In this region a mass of BAT is found under the subcutaneous WAT ^{1, 2, 3} and therefore the temperature of the skin overlying it provides a reliable and accurate readout of BAT temperature and activation. Please, note the minimal size of the scapulae in newborns, which defines a relative broader interscapular region at this life stage compared, for example, with adult humans. The exact position of the BAT area in the IRT images can be seen on the right image (**Figure 2b** of the current version of the manuscript). In all the cases, the **average temperature** of these areas was measured, as we considered it more representative of the activation of the whole BAT interscapular region. BAT temperature was not adjusted by weight or size.

Please, consider that because of **Reviewer#3** suggestion, other additional areas, namely neck and ear have been analyzed in the new version. In the case of the neck, the upper part of the nuchal fold was selected to avoid artifacts related to the skin bending; this could be achieved because each thermogenic image was associated with a visible spectrum, normal image, where anatomical boundaries/regions could be easily observed. For the ear, the outer part, the scapha, was chosen. All this information, also requested by **Reviewer#3**, has been included in the new version of the manuscript. We thank the Reviewer for this excellent comment that clearly improves our manuscript.

Reviewer's comment 2: *In general, statistical analyses and level of detail provided in Methods are adequate. However, the variability in the number of individual data shown in the correlation analysis graph for each of the hormonal parameters should be clarified.*

Response: We agree this point with the Reviewer. This information has been added to the new version. Moreover, as part of the submission package a new Excel file named **Source Data** is provided. This file contains all the raw statistical analysis of this work.

Reviewer's comment 3: *Figure 2. (A) The diagram for the cold exposure protocol is very useful. However, the control diagram, also depicting an 11-minutes bar, is somewhat misleading. (C) It should be defined how were calculated the body and BAT temperature changes at the cold exposed condition since different measures were determined (at minute 7 or at minute 11). (D-I) Data are presented in histograms. In the legend it should be better indicated as 'Data are means +/- SEM' and not 'Center values represent average; error bars represent SEM' as stated. Also check the definition of the statistical significance comparison for each of the symbols.*

Response: Following the suggestion of the Reviewer the diagram (**Figure 2a**) has been modified. In relation to **Figure 2c**, the BAT temperature changes were determined at minute 7,

subtracting BAT temperature at minute 7 from basal (minute 0) BAT temperature; this is the same for **former Figures 6d-i (Figures 7d-i in the new version)**. As requested by *Nature Communications* style, data in **Figures 2c-g and Figures 3a-f** are now represented as box plots with median, interquartile range and 10th-90th percentiles; the same style has been used in **Suppl. Figure 2a**. Finally, the definitions of statistical significance have been checked and corrected, as suggested; in this sense to avoid misunderstanding and following **Reviewer#2** comment, we have used asterisks (*, ** or ***) and lines to display the statistical comparisons.

Reviewer's comment 4: *Figure 6. Which is the difference between data in (B) and (C) (i.e., violin plots showing the individual body temperature changes at day 1 and day 2, as stated in the corresponding figure legend)? (D-I) Correlation analyses between BAT temperature change and circulating factors in cold exposed newborns at*

Response: We apologize for not being clear in these graphs. The displayed information is as follows (please, note that **Figure 6** is **Figure 7** of the new version):

- **Figure 7b:** Violin plot shows the individual body temperature changes between minute 7 and minute 3 at days 1 and 2.
- **Figure 7c:** Violin plot shows the individual body temperature changes between minute 7 and minute 0 at days 1 and 2.
- **Figures 7d-i:** the change in BAT temperature was calculated subtracting BAT temperature at minute 7 from basal (minute 0) BAT temperature.

Reviewer's comment 5: *Headings of the figure legends should be changed since no direct 'effects' were studied.*

Response: The headings have been changed accordingly.

Reviewer's comment 6: *There is a repeated reference (the reference in numbers 19 and 28 is the same).*

Response: We apologize for this mistake that has been amended in the new version

Reviewer's comment 7: *Reference 11 is not complete.*

Response: We apologize for this mistake that has been amended in the new version

We thank again the **Reviewer#2** for his/her excellent insight and comments that have improved the scope and quality of our manuscript.

REVIEWER#3

Overall comment: *The manuscript „BMP8B sets up brown fat thermogenesis in newborns” by Urisarri et al. studies early postnatal brown fat thermogenesis in humans, aiming to unravel the underlying molecular mechanisms and the possible benefits. The rationale of the study is of great importance and timely given the vast interest in brown fat at the moment. Most importantly, human data on the topic are scarce and urgently needed. The manuscript is well written, the design of the study very carefully done considering the ethical issues working with human newborns, and the data are original and of high relevance. However, there are some technical issues that need to be addressed.*

Response: We thank the Reviewer for the positive view of our manuscript and the excellent comments and suggestions. We also believe that the current manuscript provides novel and important data that enhances our understanding of the role of BAT in the regulation of temperature in neonatal stages. A detailed point-by-point response to the comments is included below.

Reviewer's comment 1: *The major problem with peripheral cold stimulation (also in adult humans) is the fact that in addition to potentially triggering BAT, it also induces a massive vasoconstriction in the extremities (as seen in Fig 2C). This lead to a shunting of the circulation and thereby potentially a maintenance of core body temperature, which could be misinterpreted as BAT thermogenesis. In the current study, body temperature was only recorded by contact infrared thermometer under the arm, which likely provides values that drop below body temperature upon cold stimulation due to reduced blood flow. Therefore, a second more independent measurement would be required, ideally rectal temperature. I understand that this might be difficult in newborns also from an ethical*

perspective, but the authors have infrared pictures available showing the ear temperature. While this might not be true inner ear temperature, it would probably still work as a proxy of body temperature that is less affected by vasoconstriction. Also, the neck region, where the skin is very thin, might be suitable. I would therefore recommend reading also the temperature data from these areas and then calculate a BAT/ear temperature value for each individual to assess BAT activation.

Response: This is a very interesting point. We have consulted with the Ethical Committee of our Hospital and it is not possible to get permission to measure rectal temperature in newborns for this study. Therefore, following Reviewer suggestion, we have made 2 new independent measurements, namely neck and ear. In the case of the neck, the upper part of the nuchal fold was selected to avoid artifacts related to the skin bending; this could be achieved because each thermogenic image was associated with a visible spectrum, normal image, where anatomical boundaries/regions could be easily observed. For the ear, the outer part, the scapha, was chosen. We would like to note that we also tried to analyze the temperature of the ear canal using the same thermographic images. However, after evaluation of the images, we concluded that measurements showed inconsistencies due to the fact that the angle of the canal (conditioned by the head position of the baby) with the camera lens influence the point we were able to measure. Thus, a perfectly aligned canal allowed to detect deeper points of the canal (with higher temperature), while non-aligned allow to detect outer parts of the canal (with lower temperature). The analysis of neck and outer ear areas provided interesting information. For example, as predicted by the Reviewer, the data showed that:

- The neck region, which is anatomically closer to the BAT interscapular area (see image on page 3; **Figure 3** from ²), shows higher basal temperature (minute 0) and a better defense to cold exposure (minutes 3, 7 and 11) that the deltoid region and the ear (**Figures 2d-f**).
- The ear, which is more peripheral, mainly cartilaginous, less irrigated, and subsequently less affected by vasoconstriction, displays a lower basal temperature and a worse defense after cold exposure at any time (minutes 3, 7 and 11) (**Figures 2d-f**).
- The **BAT/ear temperature ratio** (what an excellent suggestion!), as the Reviewer anticipated, shows that BAT is significantly activated after cold exposure at any time (minutes 3, 7 and 11) (**Figure 2g**), reinforcing the whole message of the manuscript. We extremely thank the Reviewer for suggesting us to perform this analysis that clearly improves our manuscript; it is an excellent comment, thank you.
- Moreover, to have a broader picture of our new analysis, we correlated the new analyzed temperatures with the BAT IRT measurements. Our results showed that the temperature of both the neck and the ear regions, exhibit highly significant gender-independent linear correlations with BAT temperature (**Figures 1e-j**), similarity to the deltoid temperature (**Figures 1b-1d**). This evidence reinforces the idea that BAT temperature is associated to body temperature in human newborns.

***Reviewer's comment 1:** I am not an expert in statistics, but it seems to me that the results in Fig 2d-i have a design with 2 independent factors (cold, time), so a 2-way repeated measure ANOVA would be appropriate for the analysis, which would also provide an interaction p-value that could be of interest.*

Response: We thank the Reviewer for this question. As he/she states the design reflects a 2-way repeated measures ANOVA, nonetheless the estimation of the model was made in the context of **Linear Mixed Models** ^{4,5} for two main reasons:

1. First, we decided to take advantage of the ability of these kinds of models to give unbiased results in the presence of (random) missing data.
2. Assumptions of repeated measures ANOVA were not met (absence of outliers, covariance homogeneity of between-subject factor, normality).

The fixed part of the selected model takes the following form: where Y represents the response, beta 0 the intercept, beta 1 and beta 2 the slope of the main effects, and beta 3 the slope of the interaction between treatment and day. Patient ID is included as random intercept in the model.

The model estimation is followed by a pairwise contrast of the adjusted means, and the graphical representation of the contrast for the interaction term. Please, see **Figures 3a-f** and **Suppl. Table 2**. A similar analysis was also used in the analysis of the data of Figure 2c. All the statistical details can be found in the **Source Data** Excel file.

Reviewer's comment 2: It is unclear how the BAT temperature was calculated. Was it the mean temperature of the selected area or the maximum value of the area? If it was the first case, was the area somehow adjusted to newborn weight or size?

Response: We thank the Reviewer for this comment, also raised by **Reviewer#2**, and apologize for not being clear in the first version of the manuscript. In all the cases, the average temperature of the BAT area was measured, as we considered it more representative of the activation of the whole BAT interscapular region. BAT temperature was not adjusted by weight or size, but as noted in **Table 1**, the variations in body weight and length were minimal: 3290.2 ± 51.0 g and 49.7 ± 0.3 cm, respectively.

Reviewer's comment 3: The discussion reads a bit unfocussed and extensive. It should be shortened to maintain the focus on the data provided.

Response: Following suggestion the discussion has been focused and shortened in the new version.

Reviewer's comment 4: The conclusions on the molecular mechanisms depend heavily on the quality of the ELISA used in the study. The authors should provide information on the inter- and intraassay variance of these kits (were all samples measured in one run?), as well as publications where these kits were validated. Did the authors also check TSH levels as this might be a more sensitive readout of thyroid function than total T4 and T3? It should also be added that the kits measure total hormone and not free hormone.

Response: As wrote in the methods, plasma hormonal parameters were analyzed using the commercial kits: **(i)** T4 (EIA-1781; sensitivity: 0.5 μ g/dl; intra-assay coefficient of variation (CV): 3.1-4.3% ; inter-assay CV: 2.4-4.5%; *DRG Diagnosis*; Marburg, Germany); **(ii)** T3 (EIA-1780; sensitivity: 0.2 ng/ml; intra-assay CV: 3.2-9.6%; inter-assay CV: 1.2-10.3%; *DRG Diagnosis*; Marburg, Germany); **(iii)** TSH (*AutoDELFIA® Neonatal hTSH*, B032-312; sensitivity: 2 μ U/ml; intra-assay CV: <6%; inter-assay CV:1.2-1.5%; *Perkin Elmer*; Boston, MA, US); **(iv)** FGF21 (*HLPPMAG- 57K*; sensitivity: 0.003 ng/ml; intra-assay CV: <10%; inter-assay CV: <10%; *EMD Millipore Corporation*, Billerica, MA USA) and **(v)** BMP8B (*MBS944757*; sensitivity: 2.34 pg/ml; intra-assay CV: <8%; inter-assay CV: <10%; *MyBiosource*; San Diego, CA, USA). Samples were collected from newborns along the different months which recruitment took place at the Neonatology Unit of the University of Santiago de Compostela Hospital Complex (CHUS). In order to improve our analysis potency and diminish intra-assay variability we decided to store those samples and analyze them in batches. Specifically, T4, T3, FGF21 and BMP8B measurements were performed in three different runs. We carefully keep a representative and homogeneous number of samples from all experimental groups included in the manuscript. The assay reliability of the TSH, T3 and T4 assays is beyond any doubt. These assays are used in the clinical setting both in adults, children, and neonates for the diagnosis of hypo- and hyperthyroidism. In relation to BMP8b we used a quantitative sandwich enzyme immunoassay. As expected for this kind of assay it gives us a high sensitivity (<5 pg/ml) and a wide range of concentrations to be measured. The specificity of the assay is based on the information provided by the supplier stating that the antibody does not cross-react with different BMP8B analogues. Because we were measuring human samples, we were unable to carry out a more stringent control, using samples derived from BMP8B KO mice, which is our preferred strategy. The same comments also apply to FGF21. Again this assay offers a very good sensitivity (in the region of 4 pg/ml) and the antibody appears to be specific since it does not cross-react with other FGFs (e.g. FGF19, FGF23) or with FGF analogues according to the supplier. Again, because we were measuring human samples, we were unable to carry out a more stringent control, using samples derived from FGF21 KO mice, which is our preferred strategy.

Regarding **TSH levels and thyroid hormone levels**, firstly, we thank the reviewer for calling our attention of the need to be precise regarding the measurement of thyroid hormones. We have stated now in the manuscript that hormonal measurements described were **total T3 and total T4 levels**. We apologize for this omission. We analyzed TSH serum levels on day 2, as this is a normal procedure in the Neonatology and Metabolic Diseases Units of Santiago de Compostela Hospital and proceed by doing the analogous analysis than it was done with T3 and T4. Our data showed no major changes in TSH after cold exposure and no correlation either with body or BAT temperatures.

We thank again the Reviewer#3 for his/her excellent insight and comments that have improved the scope and quality of our manuscript.

REFERENCES

1. Enerback S. Human brown adipose tissue. *Cell Metab* **11**, 248-252 (2010).
2. Lidell ME. Brown Adipose Tissue in Human Infants. *Handb Exp Pharmacol* **251**, 107-123 (2019).
3. Scheele C, Wolfrum C. Brown Adipose Crosstalk in Tissue Plasticity and Human Metabolism. *Endocr Rev* **41**, (2020).
4. Pinheiro JB, DM. *Mixed-effects models in S and S-PLUS*. Springer (2000).
5. Jiang J. *Linear and generalized linear mixed models and their applications*. Springer (2007).

Reviewer #1 (Remarks to the Author):

The study remains flawed with inadequate methodology. For example:

1. There was a 25% difference in the distance used between the subject and thermal image camera which is far too high to enable robust measurements. In addition, 3 different investigators were involved but no evidence of robust consistency is shown between each is shown. It is clear from Figure 2b there is significant movement of the infants between images which will further confound the results obtained. The delineation of putative brown fat sites is unclear, with thermal image measurements confounded by the substantial and variable amount of subcutaneous fat present in this region. No account appears to have been of this.
2. Blood sampling is poorly described from the results it is apparent not all infants were sampled, but no mention of this is made in the methods.
3. The correlations presented are implausible as it is unlikely that alleged body or brown fat temperature will be determining the plasma concentrations shown. Males and females must be considered separately. Moreover, some values are zero e.g. T4 and FGF21 which suggests sample deterioration. In addition, free T4 and T3 should have been measured as well as total hormone concentration. No details of sample processing are presented, or validation of the assays used for samples taken from neonates which in some cases seem very different from other publications.
4. The statistical analysis failed to take into account the multiple comparisons made.

Reviewer #2 (Remarks to the Author):

The authors have carefully responded to questions and criticisms. New data and changes made by the authors upon referee's suggestions have significantly improved the interpretation and discussion of the data. I consider it as an extraordinary contribution to this field of study.

Reviewer #3 (Remarks to the Author):

The authors have adequately addressed all my concerns, and I would support publication of the manuscript.

Reviewer #4 (Remarks to the Author):

General Comments

The manuscript consists of two components. The first component tests a protocol for quantifying BAT thermogenesis in neonates. Published data on changes in skin temperature in neonates is scant, and the creation of a published protocol is highly valuable to researchers investigating changes in adipose tissue and metabolism across development. It is debatable whether this component is of interest to a broad audience.

The second component of the paper examines the relationship between infant BAT thermogenesis and blood biomarkers of metabolic function. The authors document an interesting association between BAT thermogenesis and bone morphogenetic protein 8B (BMP8B).

The manuscript lacks a clear discussion of the methodological limitations of the study and how they might influence the results and interpretation.

Introduction

Consider mentioning at the end of the last paragraph of the introduction that you will explore the metabolic and endocrine correlates of infant BAT thermogenesis.

Results

I think that it would be helpful to the reader to briefly provide some information about the study sample at the beginning of the results section. How many infants were included in the study? Where did the study take place (since BAT varies across human populations)? It is not clear until

later in the Results section that the sample is divided into two groups – a control group and cold condition group. I recommend clarifying this in the first paragraph. I also recommend including a table that presents how many infants were included in each group and the sex composition of each group.

The authors state that “this evidence demonstrates for the first time that BAT temperature is associated with human body temperature in newborns”. This is technically not true since Lubowska et al. (2019) also published similar data in International Journal of Environmental Research and Public Health. Please explain why the sample sizes are so variable across the different figures in Figure 1. Also, do these relationships reflect data that were collected after the control scenario or the cold challenge scenario?

The authors hypothesize that babies that experience cold exposure on day 1 are able to maintain warmer body temperatures in response to the cold challenge on day 2. This could simply be due to developmental changes in BAT and other metabolic pathways. Do you find a similar pattern in the control group?

Discussion

It is not clear what the authors mean by a perfect correlation with body temperature in the second paragraph of the Discussion section.

At the beginning of the Discussion section, the authors state “We also demonstrate that BMP8B, but no other well-known thermogenic hormones, is likely involved in this process...” However, later they state “T4 production is clearly associated with body temperature and BAT function”. Please resolve this contradiction.

Methods

It is clear that variation in body weight, length and BMI is limited in the sample; however, body fatness is strongly associated with skin temperature. I am curious whether controlling for BMI in your analyses would impact the results.

Are there differences in weight loss between infants that experienced the cold challenge and the control infants? This would raise serious issues regarding the ethics of the protocol and is worth testing.

It is critical to note that the data was collected at a room temperature that was lower than the thermoneutral zone for infants (Hey and Katz 1970). Even the control scenario exposed the infants to a cooling stressor. This should be mentioned in the manuscript.

How were the infant blood samples collected? For instance, were they collected from a heel stick?

What is the sensitivity of the infrared contact thermometer?

The authors state that “the temperature of the skin overlying it provides a reliable and accurate readout of BAT temperature and activation.” While infrared thermal imaging may represent the most efficacious and safe option for quantifying BAT thermogenesis in infants, this is an indirect method that has limitations (for instance confounding factors such as adiposity). The authors should discuss these limitations in the Discussion section.

The description of the thermal imaging protocol is lacking key information. It is still unclear how the authors determine the average skin temperature of the interscapular region. Did they use the Box tool in FLIR tools and create an area based on particular landmarks? Without this information, it is not clear whether the measurements were captured in a standardized way. What was the emissivity setting? How were the infants positioned? Were the infants held by someone or resting on a surface?

REVIEWER#1

General comment: *The study remains flawed with inadequate methodology.*

Response: We respectfully disagree. We are quite disappointed that we have failed to convince the Reviewer#1 of the reliability of our study. We will try again to address his/her previous criticisms, as well as the new ones.

Reviewer's comment 1: *There was a 25% difference in the distance used between the subject and thermal image camera which is far too high to enable robust measurements. In addition, 3 different investigators were involved but no evidence of robust consistency is shown between each is shown. It is clear from Figure 2b there is significant movement of the infants between images which will further confound the results obtained. The delineation of putative brown fat sites is unclear, with thermal image measurements confounded by the substantial and variable amount of subcutaneous fat present in this region. No account appears to have been of this.*

Response: We thank again the Reviewer#1 for these comments. Even though, these different points were quite well discussed in the previous rebuttal, we will try to provide further evidence addressing these issues to ensure that any remaining doubt disappear.

a) **Distance:** Our data clearly and unequivocally show that distance is not a limiting factor in measuring accurately BAT temperature using thermographic methods. This is demonstrated in **Supplementary Figures 1a-b** and by Quantum Physics (**Rebuttal letter of the former version**). Ten centimeters (40-50 cm as distance range) is negligible, since we demonstrated that a much larger range (10-75 cm) does not have any impact in the IRT measurements. We would like to recall that more than 2000 images were analyzed. We honestly believe that a 10 cm range in the focus distance of >2000 images is a good level of precision. In this sense, **IRT is currently being used in massive fever screenings on people in airports and factories, as a consequence of COVID-19 pandemic**¹⁻¹¹ and has been even proposed for screening of fever in children¹². Thus, we are convinced that a variation of 10 cm is not a source of error, as being demonstrated by our experimental data and Quantum Physics approaches. In this sense, a recent review about IRT states that *"For the distances typically used in measuring BAT (<2m) the effect (of distance) will be minimal"*¹³.

b) **Number of investigators:** Besides the fact that having different investigators is mandatory for this study given that **i)** Hospital clinicians work in shifts and obviously that **ii)** we cannot plan births according to the shifts of one single clinician, having 3 different investigators doing the analysis is a **design strength**. The fact that the sampling was performed by 3 different researchers/clinicians reinforces the conclusions. Certainly, this is a key factor for randomization, which is critical in all clinical studies, and it is specifically asked in the *Reporting Summary File* in *Nature Journals*.

c) **Movements of the babies:** One strength of our study is that it can be carried out in **real living conditions**. In this context, it was desirable that the babies moved. In fact, it would be worrying if that they did not move, as this will imply that they were under restrain stress, a confounding factor we definitely wanted to avoid. This issue was considered from the very beginning by **i)** video recording of the babies when being assessed and **ii)** ensuring that the analysis was performed on pictures, which are static and therefore the areas were precisely delineated. In addition, the *E60bx Thermal FLIR Camera*, has a **60Hz frame rate which ensures the acquisition of very clear and sharp images with no distortions due to movements** (at least when the object observed is not moving at very high speeds, like, for example, a car). This is totally demonstrated looking at the images provided in the figures. They look clear and there is no sign of major shakiness or blurred areas. In our opinion, this technical point is crucial, cameras with low frame rate would generate those

typical images where the object is “moving”, but ours is generating **static images in which the analysis is optimal.**

Reviewer’s comment 2: Blood sampling is poorly described from the results it is apparent not all infants were sampled, but no mention of this is made in the methods.

Response: We agree this comment. We thank him/her for this point because it is clear for us that we did not accurately explained some details in our Methods. We deeply apologize for that. We will explain this now in detail (**page 18; red labelled**):

- Blood was extracted by venepuncture, which is the best method for extracting blood from neonates, as recommended by World Health Organization (WHO) ¹⁴, from the dorsal part of the hand using a winged steel needle (23 gauge), with an extension tube (a butterfly).
- Newborns had the following postnatal ages when sampled: the youngest one was 4 hours-old when the blood extraction was performed, while the oldest one was 46 hours-old. The average age for day 1 was 11.4 ± 0.6 hours after birth and the average of age for day 2 was 33.6 ± 0.8 hours after birth.
- The blood volume of a newborn is approximately **70 ml/Kg** ¹⁵, this means that a “standard” (3.0 Kg) human newborn has around 210 ml of blood.
- Existing guidelines, for example from WHO ^{14,15} specify that **blood volume limits** for pediatric subject should range from **1% to 4%** of the patient's total blood volume over 24 hours. Obviously, baby subjects imply a lower %.
- Therefore, for Ethical, Clinical and Scientific reasons, the amount of blood extracted for each baby was very limited. Precisely, the **volume of collected blood was 6 ml for each extraction** (3 ml for day 1 + 3 ml for day 2), corresponding to approximately 3% of the total blood volume.
- From these 3 ml of blood sample (for each time point), 1.5 ml were used for analytic routine analyses in the Neonatal Service. The **remaining 1.5 ml were used for this study.**
- **The hematocrit of a newborn infant is around 55%** ¹⁵, this means that we had 45% of serum that equals to **0.675 ml (675 µl; approximately)** per baby per time point. This was not a perfect fixed amount, for some babies we had up until 700 µl of serum, for other babies we had 600 µl of serum. The amount of serum required for the analyses is explained in **Rebuttal Table 1.**

Rebuttal Table 1: Serum volumes for the analyzed parameters

	Volume serum /reaction (µl)	Duplicated	Volume serum /assay (µl)
Glucose	5	yes	10
Triglycerides (TG)	15	yes	30
T4	25	yes	50
T3	50	yes	100
TSH	100	yes	200
FGF21	15	yes	30
BMP8B	100	yes	200
Total			630

Therefore, answering the specific question, **yes of course, all the babies included in this study were sampled.** Please, it is important to take into consideration that these were 11.4-33.6 hours-old human newborns and if, for a particular subject we did not have enough sample, that baby could not be assayed again, because of two reasons: **i)** the window period of <48 hours of life was lost and **ii)** more importantly, Ethical, Clinical and Scientific reasons were and are against of multiple blood sampling in healthy human babies. This kind of study differs in this sense, from the classical investigations performed in **i)** adult humans in which the amount of blood is

not unlimited, but evidently much more abundant than in a neonatal human infant or **ii**) preclinical models (rats, mice) that can be assayed multiple times. In this sense, the reason why in some of the analysis not all the subjects are present is basically because we did not have enough amount of sample to run 100% accurate, replicated and rigorous assays, according to the same amount of sample loaded in other experiments. In fact, this happens basically for BMP8B and TSH because the order of the analyses was: **1**) glucose → **2**) TG → **3**) T4 → **4**) T3 → **5**) FGF21 → **6**) BMP8B → **7**) TSH (requested by **Reviewer#3** for the first revised version). In summary, when we did not have enough sample to run a perfect assay for a particular sample, that sample was not assayed, this is the only reason. In this sense, despite we have already submitted a complete **Source Data File**, we will be more than happy to provide the raw, individual data for all the babies in all the conditions. Please, see also our response to **Reviewer#1/comment 3** about FGF21. We thank again the Reviewer#1 for this comment and once again, we apologize for not explaining this better in the former versions of the manuscript.

Reviewer's comment 3: *The correlations presented are implausible as it is unlikely that alleged body or brown fat temperature will be determining the plasma concentrations shown. Males and females must be considered separately. Moreover, some values are zero e.g. T4 and FGF21 which suggests sample deterioration. In addition, free T4 and T3 should have been measured as well as total hormone concentration. No details of sample processing are presented, or validation of the assays used for samples taken from neonates which in some cases seem very different from other publications.*

Response: We do not agree this comment. There are multiple physiological reasons by which BAT temperature, a sign of the extent of BAT activation, may correlate with circulating parameters, either because some of the studied hormones (e.g. BMP8B, FGF21 and THs) are known activators of BAT activity or, reciprocally, because some of them (e.g. BMP8B and FGF21) are known to be released by active BAT¹⁶⁻³³.

a) Correlation analysis: According to the *Encyclopedia of Bioinformatics and Computational Biology* (<https://www.sciencedirect.com/topics/medicine-and-dentistry/correlation-analysis>), among thousands of many other research articles, books, websites from the most prestigious Departments of Mathematics and/or Statistics of the most prestigious Schools of Mathematics and/or Statistics of the most prestigious Universities worldwide, a **Correlation Analysis** is:

“A statistical method used to evaluate the strength of relationship between two quantitative variables. A high correlation means that two or more variables have a strong relationship with each other, while a weak correlation means that the variables are hardly related. In other words, it is the process of studying the strength of that relationship with available statistical data”.

Of course, that relationship can be causal or not. Our data clearly demonstrate that there are highly significant correlations/associations/relationships between several circulating parameters and body/BAT temperatures and **that is mathematically irrefutable**. Furthermore, our data have a good degree of intervention (considering the idiosyncrasy of the studied subjects, and we emphasize this again: 1-2-day old human newborns), because we have been able to run cold-exposure experiments in newborn infants, an experimental setting that, although it may look straightforward, has logistic difficulties and challenges, but in our view, has a biological readout. To determine whether the changes in molecular markers underlie the temperature changes, it would be ideal to have the chance to run molecular mechanistic studies, as we have done in the past (and hopefully will keep going) using pre-clinical models, such as rats or mice, in which we have made some contributions to the understanding of how some hormones, such as T3, T4, FGF21 or BMP8B, among others modulate BAT thermogenesis¹⁶⁻³³. However, this study has been performed in 11.4-33.6 hours-old human newborns and again, there are Ethical, Clinical, Scientific, and clearly

common-sense, reasons that do not allow this kind of experimental approaches either in adult humans and even less in babies. Therefore, yes, we agree that we do not have a molecular mechanistic link in this study, but if the same Reviewer#1's rationale was applied to all the Clinical Literature, **a substantial number of papers already published in some of most esteemed journals** (and we will omit here the obvious Journal names) **of the Biomedical fields would be invalid**. As Scientists, we cannot believe, assume, or accept that. We thank the Reviewer#1 for this comment, but with all the kind, polite and due respect, we fully disagree with this view of our manuscript.

- b) **Sex dimorphism:** Our data clearly show that there is no sex dimorphism in the graphs displayed in **Figures 1b-j** of the former version of the manuscript. Those results show that correlations of BAT temperature with temperature of deltoid, neck and ear outer areas are similar in boys and girls. In addition to this information, we would like to state that apart from body weight, body length and cranial perimeter, none of the parameters studied: gestational age, age (day 1 and day 2), glucose, TG, T4, T3, TSH, FGF21 and BMP8B showed any significant difference between boys and girls (**Table 1 of the former version of the manuscript and Rebuttal Table 2**).

Rebuttal Table 2: Parameters in Male and Female newborns

Parameter	Males	Females
N	28	22
Gestational age (weeks)	39.7 ± 0.2	39.5 ± 0.3
Birth body weight (g)	3392.5 ± 72.0	3160.0 ± 62.5 *
Birth body length (cm)	50.3 ± 0.3	49.1 ± 0.4 *
Birth Cranial perimeter (cm)	35.0 ± 0.2	34.2 ± 0.2 **
Day 1 (hours)	11.0 ± 0.8	11.9 ± 1.0
Day 2 (hours)	33.5 ± 1.1	33.8 ± 1.1
Day 1 Glucose (mg/dl)	64.8 ± 2.0	67.4 ± 2.3
Day 2 Glucose (mg/dl)	69.3 ± 1.6	71.2 ± 2.1
Day 1 TG (mg/dl)	92.8 ± 4.0	95.2 ± 6.1
Day 2 TG (mg/dl)	126.7 ± 5.4	126.8 ± 8.6
Day 1 T4 (µg/dl)	9.0 ± 0.9	8.4 ± 1.0
Day 2 T4 (µg/dl)	8.9 ± 0.8	8.8 ± 1.0
Day 1 T3 (ng/ml)	2.6 ± 0.3	2.6 ± 0.3
Day 2 T3 (ng/ml)	2.6 ± 0.2	2.2 ± 0.3
Day 2 TSH (mUI/l)	2.1 ± 0.3	1.6 ± 0.3
Day 1 FGF21(pg/ml)	86.6 ± 23.1	130.0 ± 51.1
Day 2 FGF21(pg/ml)	125.2 ± 34.2	139.6 ± 48.2
Day 1 BMP8B (pg/ml)	628.7 ± 43.7	618.2 ± 72.7
Day 2 BMP8B (pg/ml)	776.3 ± 30.7	714.2 ± 47.5

Data are expressed as MEAN ± SEM; *P<0.05, **P<0.01 vs. Males.

Therefore, while the question of the Reviewer#1 is convenient and of real interest, we are not sure whether it would provide significant improvement on a potential revised version of the manuscript, as it would duplicate the number of panels of the paper, which currently has **90 graphs, 1 Table and 3 Supplementary Tables**. We are not sure whether a manuscript with **180 graphs, 1 Table double size of the current one and 3 Supplementary Tables double of the size of the current ones** will flow easily and it would be likely more difficult for readers (in the case that the paper is accepted) to get the main messages of the study. Certainly, we leave this at the discretion of the Editor.

- c) **Zero values: There are no zero values.** With all the due respect, this is a misinterpretation of the results. The FGF21, T3 and T4 circulating levels that we have been obtained are all the physiological range described for babies at this stage of life. **In this moment of the life, when the endocrine axes set up after the washout of maternal hormones, it is totally normal to find wide ranges in hormone concentration**³⁴⁻⁴¹. This happens for thyroid hormones, GH axis and for FGF21 (please, see next point), among others^{15,34-41}. The problem here is visual: the scale of the Y-axes: 0-25 µg/dl for T4 in **Figure 4e** (these babies are 11.4 hours old) and specially for FGF21 (0-600 pg/ml), **therefore the smallest values are closer to the X-axis, but they are not zero.**
- d) **FGF21:** It is known that the values of this hormone range from extremely low values up to > 600 pg/ml during the first hours of life (*Int J Obes (Lond) 2015 May;39(5):742-6; PMID: 25599612*)⁴². We have access to the raw data of that *IJO* paper, because one of the senior authors of that study is *Prof. Francesc Villarroya*, is also author of our current submission. In this manuscript, the FGF21 levels in the neonatal humans were presented for the first time. The values are the following (**Rebuttal Table 3**):

Rebuttal Table 3: Circulating FGF21 concentration ranges in newborns

FGF21	Age (hours)	MEAN ± SEM (pg/ml)	n	Minimal value (pg/ml)	Maximal value (pg/ml)
Our study	33.6	173.7±44.9	34	0.76	600.3
PMID: 25599612	35	212.0±54.0	11	0.0 (undetectable)	618.8

Therefore, **the values that we obtain are in the normal range of variation of this hormone at this stage of life.** Importantly, that cohort and analyses were done in another Hospital (in Barcelona) than ours (in Santiago de Compostela). Therefore, two different cohorts of babies, from two Hospitals, from two geographically distinct places show similar ranges. Notably, in most of the babies of our study, the values are higher at day 2 (33.6 hours after birth) than during day 1 (11.4 hours after birth) (**Rebuttal Figure 1**), which reinforces our conclusions. In fact, from the whole set of babies, only two of them with initial high values (364.7 pg/ml and 374.7 pg/ml, please **see graph below**) showed a decrease larger than the average of the increase. **All the babies with extremely low values on day 1 exhibited a marked increase on the day 2.** In fact, the baby with the lowest value of FGF21 at day 1 (0.076 pg/ml) was among those with the lowest postnatal life (9 hours); on day 2 his/her circulating levels raised to 22.3 pg/ml at 34 hours of postnatal life, corresponding to a 293.4x fold increase.

Therefore, **FGF21 levels are i) within the described physiological range**⁴² **and ii) they trend to increase with age**, as it happens with other hormones and endocrine axes^{15,34-41,43}. Of course, we are happy to re-assay the samples with low values, although we cannot guarantee this will work at this stage because we do not have enough amount of serum from many individuals, as explained in **comment 2**.

Rebuttal Figure 1: Individual change in circulating FGF21 concentration between day 1 and day 2

- e) **Thyroid hormones. fT3 and fT4:** There is little doubt about the reliability reported since THs were measured in the Santiago de Compostela Hospital (regional reference laboratory) with more than 40 years of experience in taking care of the neonatal congenital hypothyroidism screening program. This reliability was ensured by following the standard criteria set up by international recognized guidelines such as the CLIA regulations⁴⁴. In practical terms, we used more stringent criteria than the ones stated by standard CLIA's specifications for immunoassay performance. These include to have a low intraassay variation, usually set at less than 20%, that in our case was more stringent in terms of intra- and inter- coefficient of variation (CV) (see below, paragraph f). Furthermore, same batches of reagents were used, and each time an analysis was performed, samples from the different groups of patients were included. We have also experience of this topic in terms of basic science, since our group has made recent contributions to the field of THs and thermogenesis in pre-clinical studies in journals like *Nature Medicine*, *Cell* or *Cell Metabolism*^{16-18,20,22,45-48}. Also, we would like to point out that the values reported in this study are like the ones reported by many others, although they may be also different to some others. That is not surprising since it is widely recognized that reference values of THs are age-, gender-, ethnicity-, and method-dependent. It is also recommended that adult reference intervals should not be used in neonates and in pediatric population at a large. Hence, it is widely assumed that each institution should establish their reference values^{34,35,37,39-41,43,49-70}.

When designing the study, we had to decide which hormones to measure since, as explained, we were limited by the number/volume of samples. The formation and maturation of the newborn's hypothalamic-pituitary-thyroid (HPT) axis begins *in utero* with fetal dependence on maternal thyroid hormones early in the pregnancy. As the fetal thyroid gland begins to produce thyroid hormones in the second trimester, the reliance decreases and remains at lower levels until birth. After birth, the detachment from the placenta and the change in thermal environment leads to a rapid increase in circulating TSH in the neonate within hours, resulting in subsequent increases in T4 and T3 concentrations^{34-41,43,49-70}. **The reason we decide to measure total T4 instead of fT4 is that in this precise time of life (birth to 5 days) there are marked variation in TH levels, these variations being much higher when measuring free THs**^{34,41,50,60}. In fact, this is known since a long time ago, for example (see Erenberg et al, *Pediatrics* 53(2):211-6; PMID: 4204574)^{34,41,50,60}. The mean concentrations of T4 and fT4 increased from cord blood levels of 11.9 µg/dl and 2.9 ng/dl, respectively, to peak values of 16.2 µg/dl and 7 ng/dl by 24-48 hours of age^{34,41,50,60}. Furthermore, it has been described that in large population cohorts of newborns the normal distribution patterns are normally observed in total T4 but not in fT4,

likely due to the reliance of circulating transporting proteins^{41,60,63-65}. This may cause **underestimation of the fT4 circulating concentrations**, obtained by several routine methods, such as fT4 indexing of several immunoassay protocols⁷¹⁻¹⁰⁴. Regarding T3, although it is known that it peaks at 24h after delivery, there is not much data available about fT3. In fact, fT3 measurement is not common since, in healthy conditions, it does not provide very relevant data, as it is widely known that the main source of T3 is the local conversion of T4 by deiodination into the target tissues¹⁰⁵⁻¹⁰⁸. Moreover, although its measurement is possible, the fT3 concentrations in the serum of neonates is near the limit of detection, and the **American Thyroid Association (ATA)** discourages its use due to the unreliability of the measurement (<https://www.thyroid.org/thyroid-function-tests/>). Subsequently, due to the limited sample availability, and taking into account all these aspects, we decided that measuring total TH in this case was much more appropriate for Clinical, Scientific and Ethical reasons, in order to obtain the most reliable and relevant results possible in these precious newborn samples.

All the reasons stated above support the analysis of total T4 and T3. Those things said, in any other scenario, for example in a study involving adult humans or pre-clinical models (rats or mice), we would be more than happy to analyze fT4 and fT3 in our samples. However, in this occasion we will be totally honest: we do not have enough sample to run new assays in a significant number of subjects. For example, in the case of group cold/day 2, we had 22 babies (please, see **Figure 6e** for T4) and now we have enough serum from only 4 subjects. Therefore, with a n of 4, any conclusions obtained will lack of statistical power. We have been thinking about another possible option: to assay the levels of fT4 and fT3 in the *Whatman* papers used for the metabolic disease screening that is routinely used in our Hospital (and in any Hospital). This is a dried blood spot for cut off for TSH (for congenital hypothyroidism) screening. However, those samples were extracted after day 2 (about 42-48 hours of life) and not immediately after the cold exposure setting. Thus, the setting is not the same and we do not think that those data would be comparable to the results obtained from the blood sampling following the cold exposure protocol. Of course, if despite all the exposed Ethical, Clinical and Scientific reasons, the Editor feels that analysis of fT4 and fT3 are demanding for this study, we would be happy to try that in the *Whatman* papers used for the metabolic disease screening. In this sense, we have extensively been in contact with *Prof. Carmen Grijota-Martínez* (CSIC and Complutense University of Madrid), thyroidologist a highly recognized national expert in the determination of thyroid hormones (who has checked all our analyses), and she could try to carry out the measurement of fT4 and fT3 on these samples.

- f) **Validation of the assays:** This particular point was already discussed in the formal rebuttal (**Reviewer#3 comment**), but perhaps our response was missed by focusing only in the response to Reviewer#1. As wrote in the methods, plasma hormonal parameters were analyzed using the commercial kits: **i)** T4 (EIA-1781; sensitivity: 0.5 µg/dl; intra-assay coefficient of variation (CV): 3.1-4.3% ; inter-assay CV: 2.4-4.5%; *DRG Diagnosis*; Marburg, Germany); **ii)** T3 (EIA-1780; sensitivity: 0.2 ng/ml; intra-assay CV: 3.2-9.6%; inter-assay CV: 1.2-10.3%; *DRG Diagnosis*; Marburg, Germany); **iii)** TSH (*AutoDELFIATM Neonatal hTSH*, B032-312; sensitivity: 2 µU/ml; intra-assay CV: <6%; inter-assay CV:1.2-1.5%; *Perkin Elmer*; Boston, MA, US); **iv)** FGF21 (*HLPPMAG- 57K*; sensitivity: 0.003 ng/ml; intra-assay CV: <10%; inter-assay CV: <10%; *EMD Millipore Corporation*, Billerica, MA USA) and **v)** BMP8B (*MBS944757*; sensitivity: 2.34 pg/ml; intra-assay CV: <8%; inter-assay CV: <10%; *MyBiosource*; San Diego, CA, USA). Samples were collected from newborns along the different months which recruitment took place at the Neonatology Service of the University of Santiago de Compostela Hospital Complex (CHUS). In order to improve our analysis potency and diminish intra-assay variability we decided to store those samples and analyze them in batches. The assay reliability of the TSH, T3 and T4 assays is beyond any doubt. These assays

are used in the clinical setting both in adults, children, and neonates for the diagnosis of hypo- and hyperthyroidism. In relation to BMP8B we used a quantitative sandwich enzyme immunoassay. As expected for this kind of assay, it gives us a high sensitivity (<5 pg/ml) and a wide range of concentrations to be measured. The specificity of the assay is based on the information provided by the supplier stating that the antibody does not cross-react with different BMP8B analogues. As we were measuring human samples, we were unable to carry out a more stringent control, using samples derived from BMP8B KO mice, which is our preferred strategy. The same comments also apply to FGF21. Again, this assay offers a very good sensitivity (in the region of 4 pg/ml) and the antibody appears to be specific since it does not cross-react with other FGFs (e.g., FGF19, FGF23) or with FGF analogues according to the supplier. Again, because we were measuring human samples, we were unable to carry out a more stringent control, using samples derived from FGF21 KO mice, which is our preferred strategy.

Reviewer's comment 4: *The statistical analysis failed to take into account the multiple comparisons made.*

Response: We do not agree with Reviewer 1 on this comment. We do not really understand it because all the multiple comparisons are present in **Supplementary Table 1, Supplementary Table 2 and in the Source Data File** provided. As stated in the previous Rebuttal, the estimation of the model was made in the context of **Linear Mixed Models**^{109,110} for two main reasons:

1. First, we decided to take advantage of the ability of these kinds of models to give unbiased results in the presence of (random) missing data.
2. Assumptions of repeated measures ANOVA were not met (absence of outliers, covariance homogeneity of between-subject factor, normality).

The fixed part of the selected model takes the following form: where Y represents the response, beta 0 the intercept, beta 1 and beta 2 the slope of the main effects, and beta 3 the slope of the interaction between treatment and day. Patient ID is included as random intercept in the model. The model estimation is followed by a pairwise contrast of the adjusted means, and the graphical representation of the contrast for the interaction term. Please, see **Figures 3a-f** and **Supplementary Table 2**. A similar analysis was also used in the analysis of the data of **Figure 2c**. All the statistical details can be found in the **Source Data File**.

REVIEWER#2

General comment: *The authors have carefully responded to questions and criticisms. New data and changes made by the authors upon referee's suggestions have significantly improved the interpretation and discussion of the data. I consider it as an extraordinary contribution to this field of study.*

Response: We thank very much the Reviewer for the positive view of our manuscript and his/her words on our work. We also thank the Reviewer for his/her comments which highly improved the quality of our study.

REVIEWER#3

General comment: *The authors have adequately addressed all my concerns, and I would support publication of the manuscript.*

Response: We thank very much the Reviewer for the positive view of our manuscript and his/her words on our work. We also thank the Reviewer for his/her comments which highly improved the quality of our study.

REVIEWER#4

General comments: The manuscript consists of two components. The first component tests a protocol for quantifying BAT thermogenesis in neonates. Published data on changes in skin temperature in neonates is scant, and the creation of a published protocol is highly valuable to researchers investigating changes in adipose tissue and metabolism across development. It is debatable whether this component is of interest to a broad audience.

The second component of the paper examines the relationship between infant BAT thermogenesis and blood biomarkers of metabolic function. The authors document an interesting association between BAT thermogenesis and bone morphogenetic protein 8B (BMP8B).

The manuscript lacks a clear discussion of the methodological limitations of the study and how they might influence the results and interpretation.

Response: We thank the Reviewer for the positive view of our manuscript and the outstanding comments and suggestions. We also believe that the current manuscript provides novel and important data that enhances our understanding of the role of BAT in the regulation of temperature in neonatal stages. A detailed point-by-point response to the comments is included below.

Reviewer's comment 1: Introduction. Consider mentioning at the end of the last paragraph of the introduction that you will explore the metabolic and endocrine correlates of infant BAT thermogenesis.

Response: We thank the Reviewer for this comment, we have introduced this suggestion in the Introduction (page 4; red labelled).

Reviewer's comment 2: Results. I think that it would be helpful to the reader to briefly provide some information about the study sample at the beginning of the results section. How many infants were included in the study? Where did the study take place (since BAT varies across human populations)? It is not clear until later in the Results section that the sample is divided into two groups – a control group and cold condition group. I recommend clarifying this in the first paragraph. I also recommend including a table that presents how many infants were included in each group and the sex composition of each group.

Response: We thank the Reviewer#4 for this comment. We agree that the information was a little bit scarce in the former version and this has been improved in the new one. In relation to the requested Table, this information was already part of the submission (referred as **Table 1** on **Methods Section/page 13/paragraph 1/lane 4**), we apologize if it was not clearly referenced, this has been improved in the new version.

Rebuttal Table 4: Parameters in Male and Female newborns (Table 1 of the manuscript)

	Males	Females	Males & Females
N	28	22	50
Gestational age at birth (weeks)	39.7 ± 0.2	39.5 ± 0.3	39.6 ± 0.2
Birth body weight (g)	3392.5 ± 72.0	3160.0 ± 62.5*	3290.2 ± 51.0
Birth body length (cm)	50.3 ± 0.3	49.1 ± 0.4*	49.7 ± 0.3
Birth cranial perimeter (cm)	35.0 ± 0.2	34.2 ± 0.2**	34.7 ± 0.1
Age day 1 (h)	11.0 ± 0.8	11.9 ± 1.0	11.4 ± 0.6
Age day 2 (h)	33.5 ± 1.1	33.8 ± 1.1	33.6 ± 0.8
Body weight change (age 1-age 2; g)	-163.5 ± 12.6	-165.2 ± 16.2	-164.3 ± 10.0

Data are expressed as MEAN ± SEM; *P<0.05, **P<0.01 vs. Males.

Reviewer's comment 3: The authors state that "this evidence demonstrates for the first time that BAT temperature is associated with human body temperature in newborns". This is technically not true since Lubowska et al. (2019) also published similar data in International Journal of Environmental Research and Public Health. Please explain why the sample sizes are so variable across the different figures in Figure 1. Also, do these relationships reflect data that were collected after the control scenario or the cold challenge scenario?

Response: We deeply apologize for not quoting that paper, which has been highly referenced and discussed in the new version¹¹¹.

In relation to the variations in the sample size in the different panels of **Figure 1**, the explanation is as follows. In the first version of the manuscript (*NCOMMS-20-24112-T*), we showed data on the correlation between body temperature and BAT temperature (in both control and cold scenarios) in Male + Female, Male and Female groups (**Figures 1b-d** of *NCOMMS-20-24112-T* and also of the first revised version *NCOMMS-20-24112A*). After the revision of the first version, **Reviewer#3** made a very interesting comment and suggestion:

“Reviewer’s comment 1: The major problem with peripheral cold stimulation (also in adult humans) is the fact that in addition to potentially triggering BAT, it also induces a massive vasoconstriction in the extremities (as seen in Fig 2C). This leads to a shunting of the circulation and thereby potentially a maintenance of core body temperature, which could be misinterpreted as BAT thermogenesis. In the current study, body temperature was only recorded by contact infrared thermometer under the arm, which likely provides values that drop below body temperature upon cold stimulation due to reduced blood flow. Therefore, a second more independent measurement would be required, ideally rectal temperature. I understand that this might be difficult in newborns also from an ethical perspective, but the authors have infrared pictures available showing the ear temperature. While this might not be true inner ear temperature, it would probably still work as a proxy of body temperature that is less affected by vasoconstriction. Also, the neck region, where the skin is very thin, might be suitable. I would therefore recommend reading also the temperature data from these areas and then calculate a BAT/ear temperature value for each individual to assess BAT activation.”

Our response to that comment was, literally copied and pasted from our **Rebuttal Letter**:

“Response: This is a very interesting point. We have consulted with the Ethical Committee of our Hospital and it is not possible to get permission to measure rectal temperature in newborns for this study. Therefore, following Reviewer suggestion, we have made 2 new independent measurements, namely neck and ear. In the case of the neck, the upper part of the nuchal fold was selected to avoid artifacts related to the skin bending; this could be achieved because each thermogenic image was associated with a visible spectrum, normal image, where anatomical boundaries/regions could be easily observed. For the ear, the outer part, the scapha, was chosen. We would like to note that we also tried to analyze the temperature of the ear canal using the same thermographic images. However, after evaluation of the images, we concluded that measurements showed inconsistencies due to the fact that the angle of the canal (conditioned by the head position of the baby) with the camera lens influence the point we were able to measure. Thus, a perfectly aligned canal allowed to detect deeper points of the canal (with higher temperature), while non-aligned allow to detect outer parts of the canal (with lower temperature)...”

Therefore, following the **Reviewer#3’s** suggestion, we re-analyzed the ear and neck temperatures in the same thermographic images where the BAT area had been previously assayed. In the case of the neck, the upper part of the nuchal fold was selected to avoid artifacts related to the skin bending; this could be achieved because each thermogenic image was associated with a visible spectrum, normal image, where anatomical boundaries/regions could be easily observed. For the ear, the outer part, the scapha, was chosen. However, given the position of the head and the babies, the ear and neck areas were not evident in all the images. Thus, **we only measured the pictures in which the measurement was reliable**, and we were totally sure that ear and neck are clearly defined. This is the reason why the sample size is variable.

Reviewer's comment 4: *The authors hypothesize that babies that experience cold exposure on day 1 are able to maintain warmer body temperatures in response to the cold challenge on day 2. This could simply be due to developmental changes in BAT and other metabolic pathways. Do you find a similar pattern in the control group?*

Response: This is a convenient question. Control babies did not show the adaptation to the cold stimulus that was observed in the exposed group. In this group, after undressing the infants on day 1, their temperature fell of 0.36 ± 0.04 °C (Time 0 min: 36.56 ± 0.7 °C vs. Time 3 min: 36.20 ± 0.06 °C; $P < 0.001$). On day 2, the body temperature of the same newborns fell of -0.40 ± 0.06 °C (Time 0 min: 36.6 ± 0.1 °C vs. Time 3 min: 36.2 ± 0.1 °C; $P < 0.01$). Therefore, the babies suffered a similar temperature decrease on both days (-0.36 ± 0.04 °C vs. -0.40 ± 0.06 °C; $P = 0.4112$; **Rebuttal Figure 2 = Supplementary Figure 6a** in the new version of the manuscript) indicating that time or any associated metabolic/developmental changes are not responsible of the adaptative response observed in cold exposed babies (**Figure 7a**). We thank the Reviewer for this interesting and relevant suggestion.

Rebuttal Figure 2: Body temperature changes at day 1 and day 2 in cold exposed newborns. Box plots display median, interquartile range and 10th-90th percentiles. Number of newborns/group: 23-24. Statistical significance was determined by Mann-Whitney test. Source data are provided as a Source Data file

(**Supplementary Figure 6** in the new version of the manuscript)

Reviewer's comment 5: *Discussion. It is not clear what the authors mean by a perfect correlation with body temperature in the second paragraph of the Discussion section.*

Response: We mean a highly significant correlation, but of course our expression was not correct since a perfect correlation would imply $r = 1$ (for a perfect positive correlation) or $r = -1$ (for a perfect negative correlation), which was not the case. We apologize for this mistake that has been amended in the new version.

Reviewer's comment 6: *At the beginning of the Discussion section, the authors state "We also demonstrate that BMP8B, but no other well-known thermogenic hormones, is likely involved in this process..." However, later they state "T4 production is clearly associated with body temperature and BAT function". Please resolve this contradiction.*

Response: In this point, we were discussing the data showed in **Figure 6e-f**, in which there is a correlation ($P = 0.01$) between circulating T4 and body temperature and a trend between T4 and BAT temperature ($P = 0.066$) at postnatal day 2 (33.6 ± 0.8 hours after birth) in cold exposed newborns. We totally agree with the Reviewer#4 suggestion in the fact that: **i)** the dispersion of the values and the significance level are not as high that in the case of BMP8B (**Figure 6k-l**) and notably **ii)** (and this is important) the variation of BAT temperature does not correlate with T4 (**Figure 6f**) as it does with BMP8B (**Figure 6i**), therefore, we have toned down and re-written that conclusion as "... our data might suggest that T4 production could be associated with thermogenesis in newborns although not as much as BMP8B" (**page 13; red labelled**). We thank the Reviewer for this excellent comment that clearly improves our manuscript.

Reviewer's comment 7: *Methods. It is clear that variation in body weight, length and BMI is limited in the sample; however, body fatness is strongly associated with skin temperature. I curious whether controlling for BMI in your analyses would impact the results.*

Response: This is a question that has a controversial interest. In fact, we doubt when we introduced BMI as part of the parameters in **Table 1**. The main reason is that BMI can provide an indication about the homogeneity of the compared groups (control and cold exposed) but it

does not really offer a reliable information about the adiposity degree in newborns, basically for two reasons: **i)** neonatal human babies have a very limited amount of adipose depots (essentially BAT is the most abundant depot that they have) ¹¹²⁻¹¹⁴ and **ii)** the weight of the head is massive in relation to the body weight: in a baby of 3,500 g, the head weights around 800 g (23% of total body weight, with the brain weighting around 400 g) ¹⁵. In contrast in an adult human of 80 kg, the weight of the head is around 6 kg (7.5 %) ¹⁵. The most accurate and used measure used to assess the growth of the baby are the body weight percentiles ¹⁵, since the use of BMI is specifically discouraged by the WHO statement "Use of the BMI-for-age growth chart is not recommended for children younger than age two years" (Center Disease Control and Prevention: <https://www.cdc.gov/nccdphp/dnpao/growthcharts/who/using/index.htm>).

In any case, following the **Reviewer#4** suggestion, we have studied the possible associations between body weight and BMI with body and BAT temperatures (**Rebuttal Figure 3 = Supplementary Figure 4** in the new version of the manuscript). Our data show that neither body weight nor BMI correlate with temperature in any of the analyzed settings/ages.

Rebuttal Figure 3:
Correlations between body or BAT temperatures and body weight or BMI
 Correlation analyses between body temperature (**a, c, e, g, i and k**) and BAT temperature (**b, d, f, h, j and l**) in newborns in control conditions at postnatal day 1 (**a-d**), postnatal day 2 in control conditions (**e-h**) and postnatal day under cold exposed conditions (**i-l**). Number of newborns: (**a**) 50, (**b**) 49, (**c**) 50, (**d**) 49, (**e**) 21, (**f**) 21, (**g**) 21, (**h**) 21, (**i**) 23, (**j**) 22, (**k**) 23, (**l**) 22. Linear regression performed by Pearson's test or Spearman's test. Source data are provided as a Source Data file.

(**Supplementary Figure 4** in the new version of the manuscript)

Reviewer's comment 8: Are there differences in weight loss between infants that experienced the cold challenge and the control infants? This would raise serious issues regarding the ethics of the protocol and is worth testing.

Response: We thank the **Reviewer#4** for bringing this interesting point. After being born, it is totally normal for the babies to lose weight in the following days¹⁵. This is what is known as **neonatal physiological weight loss** ranges between 4% and 7% of the newborn's weight, with a maximum of 10%¹⁵. It may take them up until 10 days to recover the initial body mass¹⁵. The loss of body weight is due to different reasons, such as fluid loss and the expulsion of meconium, the earliest stool of a mammalian newborns, including humans¹⁵. Unlike later feces, meconium is composed of materials ingested during the time the infant spends in the uterus: intestinal epithelial cells, lanugo, mucus, amniotic fluid, bile, and water. In our study, **all the babies analyzed, every single one, lost body weight independently of gender or group and no differences were found between control and cold exposed infants (Rebuttal Table 5 = Supplementary Table 1 in the new version of the manuscript and Rebuttal Figure 4 = Supplementary Figure 2 in the new version of the manuscript)** within the physiological range (minimal 1.2% to maximal 8.6%).

Rebuttal Table 5: Body weight change (age 1-age 2)
(Supplementary Table 1 in the new version of the manuscript)

	Body weight change (age 1-age 2; g)	Body weight change (age 1-age 2; %)
Male Control	-164.4 ± 14.8	-5.1 ± 0.4
Male Cold exposed	-162.9 ± 18.8	-4.8 ± 0.5
Male (all)	-163.5 ± 12.6	-4.9 ± 0.4
Female Control	-175.0 ± 21.4	-5.4 ± 0.6
Female Cold exposed	-152.2 ± 25.4	-4.9 ± 0.8
Female (all)	-165.2 ± 16.2	-5.2 ± 0.5
Control (all)	-170.5 ± 13.5	-5.3 ± 0.4
Cold exposed (all)	-158.7 ± 15.4	-4.8 ± 0.4

Data are expressed as MEAN ± SEM; all the comparisons are non-significant

Rebuttal Figure 4:
Individual body weight evolution between day 1 and day 2.

(a) All newborns
(b) Control newborns
(c) Cold exposed newborns.

(Supplementary Figure 2 in the new version of the manuscript)

Reviewer's comment 9: *It is critical to note that the data was collected at a room temperature that was lower than the thermoneutral zone for infants (Hey and Katz 1970). Even the control scenario exposed the infants to a cooling stressor. This should be mentioned in the manuscript.*

Response: This is a very interesting point, thank you for the suggestion. After being in a relatively stable thermoneutral uterus for the whole of pregnancy, the newborn baby enters a cooling environment and might suffer significant heat loss and hypothermia during the first hours of life. Thermoneutral zone in babies has a higher range (32-35°C) than in adult humans (26-28°C at 50% of relative humidity), since babies lack of good insulation and the heat losses by radiation, convection, evaporation and conduction are much higher^{15,111,115-131}. However, it is important to maintain a balance between an optimal thermic environment and hyperthermia because the maximal tolerable air temperature for baby is about 37°C and hyperthermic babies may have even a worse short-term outcome^{15,111,115-123}. For these reasons, the temperature management in the delivery and maternity rooms of the Neonatal Services/Units has been always a matter of debate in the literature, scaled even to the WHO, which recommends range of temperature of 25–28 °C to reduce the incidence of neonatal hypothermia but also for avoiding hyperthermia (https://apps.who.int/iris/bitstream/handle/10665/63986/WHO_RHT_MSM_97.2.pdf;sequence%3D1)^{15,120}. In our Hospital, as in many others, the temperature was set at 26-28 °C in the delivery room and at 25-26°C at the maternity area, both within the range suggested by WHO. Therefore, although the Reviewer#4 is right about the possible impact of the room temperature when undressing the babies, Ethical, Clinical and Scientific reasons support that environment. Those things said, this is an important point that we have discussed in the manuscript (**pages 12-13; red labelled**). We thank the Reviewer for this excellent suggestion.

Reviewer's comment 10: *How were the infant blood samples collected? For instance, were they collected from a heel stick?*

Response: The blood was extracted by qualified nursing staff of the Neonatal Service of the University Hospital of Santiago de Compostela, following existing guidelines, such as those from WHO^{14,15}. Blood was extracted by venepuncture, which is the best method for extracting blood from neonates^{14,15}, from the dorsal part of the hand using a winged steel needle (23 gauge), with an extension tube (a butterfly). The amount of blood extracted for each baby was very limited. Precisely, the volume of blood was 6 ml for each extraction (3 ml for day 1 + 3 ml for day 2), which was about 3% of the total blood volume. From this 3 ml of blood sample for each time point, 1.5 ml were used for analytic routine analyses in the Neonatal Service, thus the remaining 1.5 ml was given to us for this study. The hematocrit of a newborn infant is around 55%¹⁵, this means that we had 45% of serum that equals to 0.675 ml (675 µl; approximately) per baby per time point. This was not a perfect fixed amount, for some babies we had up until 700 µl of serum, for other babies we had 600 µl of serum. This information has been added to the new version of the manuscript (**page 18; red labelled**). We thank the Reviewer for this excellent comment that clearly improves our manuscript.

Reviewer's comment 11: *What is the sensitivity of the infrared contact thermometer?*

Response: Peripheral body temperature was measured using an infrared contact thermometer (*Intersurgical*; Madrid, Spain) with the following characteristics:

- Measurement distance: 3-5 cm
- Measuring Range:
 - Body: 32.0~42.9°C (89.6~109.2°F)
 - Scan: 0~50°C (32~122°F)
- Accuracy: ±0.2°C
- Resolution: 0.1°C

Reviewer's comment 12: The authors state that “the temperature of the skin overlying it provides a reliable and accurate readout of BAT temperature and activation.” While infrared thermal imaging may represent the most efficacious and safe option for quantifying BAT thermogenesis in infants, this is an indirect method that has limitations (for instance confounding factors such as adiposity). The authors should discuss these limitations in the Discussion section.

Response: The Reviewer is totally right, the limitations of IRT have been discussed in the new version of the manuscript (page 11 and pages 12-13; red labelled). We totally agree that even IRT being a good surrogate for the estimation of BAT temperature, layer subcutaneous fat insulation may vary in its thickness and therefore this could affect IRT of BAT¹³²⁻¹³⁷. In fact, in adult humans IRT analysis shows good correlations with BAT activity in lean but not obese subjects¹³²⁻¹³⁷. However, it must be taken account that white fat deposits in newborns are scarce, skin is very thin¹⁵ and the advantages of the method: **i)** non-invasive approach, **ii)** sensitivity to cold simulation, **iii)** lack of radiation exposure and **iv)** control of variables (in this study: age, weight and room temperature) make IRT likely the safest method for the investigation of BAT function in such a sensible and vulnerable human population, as newborn babies^{135,136}. In this sense, IRT has been validated and compared with PET-CT using ¹⁸F-FDG and concluded that IRT is a convenient technique for studying BAT function^{13,133,134}.

Reviewer's comment 13: The description of the thermal imaging protocol is lacking key information. It is still unclear how the authors determine the average skin temperature of the interscapular region. Did they use the Box tool in FLIR tools and create an area based on particular landmarks? Without this information, it is not clear whether the measurements were captured in a standardized way. What was the emissivity setting? How were the infants positioned? Were the infants held by someone or resting on a surface?

Response: The area of analysis was landmarked using the Box Tool in the *FLIR Tools Software*, the size of that area and the landmarks were similar for all the babies. The interscapular region was defined as the one between the scapulae or shoulder blades, an area where BAT is abundant in newborns¹¹²⁻¹¹⁴. This anatomical area is well-defined in the magnetic resonance image at the left (Figure 3 from¹¹³), where sagittal (a), coronal (b) and axial (c)

planes are shown and the interscapular BAT is highlighted in green (scale bar: 5 cm). In this region a mass of BAT is found under the subcutaneous WAT¹¹²⁻¹¹⁴ and therefore the temperature of the skin overlying it provides a reliable and accurate readout of BAT temperature and activation. Please, note the minimal size of the scapulae in newborns, which defines a relative broader interscapular region at this life stage compared, for example, with adult humans. The exact position of the BAT area in the IRT images can be seen on the right image. In all the cases, the average temperature of these areas was measured, as we considered it to be more representative of the activation of the whole BAT interscapular region. BAT temperature was not adjusted by weight or size.

In relation to the **emissivity**, this is also an interesting question from the Reviewer. Human skin emissivity (ϵ) is a current topic of study and different values have been reported, although it is certain that human emissivity is very high, nearly like a black body ($\epsilon=1$). The values that have been reported for the emissivity of the human skin range from 0.91 to 0.99^{138,139}, with a consensus of $\epsilon = 0.98 \pm 0.01$ ^{2,9,13,138-158} (<https://www.thermoworks.com/emissivity-table>; <https://www.flir.com/discover/professional-tools/how-does-emissivity-affect-thermal-imaging/>), when quoted, because in fact not many reports specify the parameter in the current literature. However, this consensus value, in our view has a potential weakness in the case of the subjects, namely human newborns, of our study. The value of $\epsilon = 0.98$ is based in multitude of studies that have analyzed emissivity in adult humans, mainly male and body areas, such as fingers, back and palm of the hand, wrist, forearm, toes and forehead. Notably, those areas with thicker skin, such as hands, outer wrist, dorsal forearm and feet show higher emissivity (in the range of 0.06) that those other areas with thinner skin, such as cheek, inner wrist and volar forearms^{2,9,13,138-158}. Moreover, recent data have also shown that women, who has slimmer skin compared to men, have an emissivity 0.046 (in average) lower than men¹³⁹. For these reasons, given the thin skin of human newborns, particularly in the interscapular area^{15,159} and despite that the $\epsilon = 0.98$ has been used before in other neonatal studies¹⁶⁰^{135,136,152,161,162}, we have considered a lower value of $\epsilon = 0.95$, which still in the range of human skin emissivity. In any case, as we extensively tried different values of emissivity when we optimized our protocols, we have also done our analyses with $\epsilon = 0.98$. As it can be seen in **Rebuttal Figure 5**, the obtained data/correlations are similar, and the conclusions of the study are exactly the same. This has been explained in the Methods Section (pages 16-17; **red labelled**).

Rebuttal Figure 5: Data obtained with $\epsilon = 0.98$

(a-c) Correlations between BAT and body temperature in newborns. **Compare with Figures 1b-d of the manuscript**

(d) Body and BAT temperature changes at control and cold exposed conditions. **Compare with Figure 2c of the manuscript**

(e) Correlation between BAT temperature and BMP8B in cold exposed 2-day old newborns. **Compare with Figure 6I of the manuscript**

(f) Correlation between BAT temperature change and BMP8B in cold exposed 2-day old newborns. **Compare with Figure 7i of the manuscript**

Number of newborns/group: exactly the same used in the Figures specified above. Data expressed as mean \pm SEM.

Finally, babies were placed in a **prone position**, as shown in **Figures 1a and 2b** of the manuscript and in the image above. They were not held by anyone; they were placed on a flat mattress.

We thank again the **Reviewer#4** for his/her excellent insight and comments that have improved the scope and quality of our manuscript.

REFERENCES

1. Buoitte Stella, A., Manganotti, P., Furlanis, G., Accardo, A. & Ajcevic, M. Return to school in the COVID-19 era: considerations for temperature measurement. *J Med Eng Technol* **44**, 468-471 (2020).
2. Charlton, M., et al. The effect of constitutive pigmentation on the measured emissivity of human skin. *PLoS One* **15**, e0241843 (2020).
3. Chetty, T., Daniels, B.B., Ngandu, N.K. & Goga, A. A rapid review of the effectiveness of screening practices at airports, land borders and ports to reduce the transmission of respiratory infectious diseases such as COVID-19. *S Afr Med J* **110**, 1105-1109 (2020).
4. Gefen, A. Infrared thermography, COVID-19 and pressure ulcer risk. *J Wound Care* **29**, 483-484 (2020).
5. Mekjavic, I.B. & Tipton, M.J. Myths and methodologies: Degrees of freedom - limitations of infrared thermographic screening for Covid-19 and other infections. *Exp Physiol* (2020).
6. Pannu, J. Nonpharmaceutical Measures for Pandemic Influenza in Nonhealthcare Settings-International Travel-Related Measures. *Emerg Infect Dis* **26**(2020).
7. Scarano, A., Inchingolo, F. & Lorusso, F. Facial Skin Temperature and Discomfort When Wearing Protective Face Masks: Thermal Infrared Imaging Evaluation and Hands Moving the Mask. *Int J Environ Res Public Health* **17**(2020).
8. Zhou, Y., et al. Clinical evaluation of fever-screening thermography: impact of consensus guidelines and facial measurement location. *J Biomed Opt* **25**(2020).
9. Dell'Isola, G.B., Cosentini, E., Canale, L., Ficco, G. & Dell'Isola, M. Noncontact Body Temperature Measurement: Uncertainty Evaluation and Screening Decision Rule to Prevent the Spread of COVID-19. *Sensors (Basel)* **21**(2021).
10. Martinez-Jimenez, M.A., et al. Diagnostic accuracy of infrared thermal imaging for detecting COVID-19 infection in minimally symptomatic patients. *Eur J Clin Invest* **51**, e13474 (2021).
11. Muller, D., Ehlen, A. & Valeske, B. Convolutional Neural Networks for Semantic Segmentation as a Tool for Multiclass Face Analysis in Thermal Infrared. *J Nondestr Eval* **40**, 9 (2021).
12. Selent, M.U., et al. Mass screening for fever in children: a comparison of 3 infrared thermal detection systems. *Pediatr Emerg Care* **29**, 305-313 (2013).
13. Law, J., et al. The use of infrared thermography in the measurement and characterization of brown adipose tissue activation. *Temperature (Austin)* **5**, 147-161 (2018).
14. Organization, W.H. *WHO Guidelines on Drawing Blood: Best Practices in Phlebotomy*, (Geneva, 2010).
15. Rennie, M.J., Robertson, NRC. *Rennie & Robertson's Textbook of Neonatology*, (Elsevier, 2014).
16. López, M., et al. Hypothalamic AMPK and fatty acid metabolism mediate thyroid regulation of energy balance. *Nat. Med* **16**, 1001-1008 (2010).
17. Whittle, A.J., et al. Bmp8b increases brown adipose tissue thermogenesis through both central and peripheral actions. *Cell* **149**, 871-885 (2012).
18. Alvarez-Crespo, M., et al. Essential role of UCP1 modulating the central effects of thyroid hormones on energy balance. *Mol. Metab* **5**, 271-282 (2016).
19. Martins, L., et al. A Functional Link between AMPK and Orexin Mediates the Effect of BMP8B on Energy Balance. *Cell Rep* **16**, 2231-2242 (2016).
20. Martínez-Sánchez, N., et al. Hypothalamic AMPK-ER stress-JNK1 axis mediates the central actions of thyroid hormones on energy balance. *Cell Metab* **26**, 212-229 (2017).
21. Seoane-Collazo, P., et al. SF1-Specific AMPKalpha1 Deletion Protects Against Diet-Induced Obesity. *Diabetes* **67**, 2213-2226 (2018).
22. Martinez-Sanchez, N., et al. Thyroid hormones induce browning of white fat. *J. Endocrinol* **232**, 351-362 (2017).
23. Gonzalez-Garcia, I., et al. Estradiol Regulates Energy Balance by Ameliorating Hypothalamic Ceramide-Induced ER Stress. *Cell Rep* **25**, 413-423 (2018).

24. Hondares, E., et al. Hepatic FGF21 expression is induced at birth via PPARalpha in response to milk intake and contributes to thermogenic activation of neonatal brown fat. *Cell Metab* **11**, 206-212 (2010).
25. Hondares, E., et al. Thermogenic activation induces FGF21 expression and release in brown adipose tissue. *J. Biol. Chem* **286**, 12983-12990 (2011).
26. Diaz-Delfin, J., et al. TNF-alpha represses beta-Klotho expression and impairs FGF21 action in adipose cells: involvement of JNK1 in the FGF21 pathway. *Endocrinology* **153**, 4238-4245 (2012).
27. De Sousa-Coelho, A.L., et al. FGF21 mediates the lipid metabolism response to amino acid starvation. *J Lipid Res* **54**, 1786-1797 (2013).
28. Hondares, E., et al. Fibroblast growth factor-21 is expressed in neonatal and pheochromocytoma-induced adult human brown adipose tissue. *Metabolism* **63**, 312-317 (2014).
29. Giralt, M., Gavalda-Navarro, A. & Villarroya, F. Fibroblast growth factor-21, energy balance and obesity. *Mol Cell Endocrinol* **418 Pt 1**, 66-73 (2015).
30. Cereijo, R., Giralt, M. & Villarroya, F. Thermogenic brown and beige/brite adipogenesis in humans. *Ann Med* **47**, 169-177 (2015).
31. Quesada-Lopez, T., et al. The lipid sensor GPR120 promotes brown fat activation and FGF21 release from adipocytes. *Nat Commun* **7**, 13479 (2016).
32. Giralt, M. & Villarroya, F. Mitochondrial Uncoupling and the Regulation of Glucose Homeostasis. *Curr Diabetes Rev* **13**, 386-394 (2017).
33. Quesada-Lopez, T., et al. GPR120 controls neonatal brown adipose tissue thermogenic induction. *Am J Physiol Endocrinol Metab* **317**, E742-E750 (2019).
34. Erenberg, A., Phelps, D.L., Lam, R. & Fisher, D.A. Total and free thyroid hormone concentrations in the neonatal period. *Pediatrics* **53**, 211-216 (1974).
35. Fisher, D.A. Thyroid function in the fetus and newborn. *Med Clin North Am* **59**, 1099-1107 (1975).
36. Nathanielsz, P.W. Thyroid function in the fetus and newborn mammal. *Br Med Bull* **31**, 51-56 (1975).
37. Schindler, A.M. Abnormal thyroid function in a healthy newborn. *Hosp Pract (Off Ed)* **22**, 246-248 (1987).
38. Balabolkin, M.I., Selishcheva, R.F. & Petruk, S.N. [Thyroid function of low birth weight newborn infants]. *Pediatrriia*, 14-22 (1988).
39. Rapaport, R. Thyroid function in the very low birth weight newborn: rescreen or reevaluate? *J Pediatr* **140**, 287-289 (2002).
40. Shih, J.L. & Agus, M.S. Thyroid function in the critically ill newborn and child. *Curr Opin Pediatr* **21**, 536-540 (2009).
41. Eng, L. & Lam, L. Thyroid Function During the Fetal and Neonatal Periods. *Neoreviews* **21**, e30-e36 (2020).
42. Sanchez-Infantes, D., et al. Circulating FGF19 and FGF21 surge in early infancy from infra- to supra-adult concentrations. *Int J Obes (Lond)* **39**, 742-746 (2015).
43. Segni, M. *Disorders of the Thyroid Gland in Infancy, Childhood and Adolescence*, (MDText.com, 2017).
44. Centers for Disease, C., Prevention Centers for, M. & Medicaid Services, H.H.S. Medicare, Medicaid, and CLIA programs; laboratory requirements relating to quality systems and certain personnel qualifications. Final rule. *Fed Regist* **68**, 3639-3714 (2003).
45. López, M., Seoane, L., Señarís, R. & Diéguez, C. Prepro-orexin mRNA levels in the rat hypothalamus, and orexin receptors mRNA levels in the rat hypothalamus and adrenal gland are not influenced by the thyroid status. *Neurosci. Lett* **300**, 171-175 (2001).
46. Varela, L., et al. Hypothalamic mTOR pathway mediates thyroid hormone-induced hyperphagia in hyperthyroidism. *J. Pathol* **227**, 209-222 (2012).
47. Gonzalez, C.R., et al. Hyperthyroidism differentially regulates neuropeptide S system in the rat brain. *Brain Res* **1450**, 40-48 (2012).
48. Álvarez-Crespo, M., et al. The orexigenic effect of orexin-A revisited: dependence of an intact growth hormone axis. *Endocrinology* **154**, 3589-3598 (2013).
49. Lemarchand-Beraud, T., Genazzani, A.R., Bagnoli, F. & Casoli, M. Thyroid function in the premature and the full term newborn. *Acta Endocrinol (Copenh)* **70**, 445-453 (1972).
50. Abuid, J., Stinson, D.A. & Larsen, P.R. Serum triiodothyronine and thyroxine in the neonate and the acute increases in these hormones following delivery. *J Clin Invest* **52**, 1195-1199 (1973).
51. Homoki, J., et al. Thyroid function in term newborn infants with congenital goiter. *J Pediatr* **86**, 753-758 (1975).
52. Chen, S.H. Thyroid function in Chinese newborn infants. *Taiwan Yi Xue Hui Za Zhi* **79**, 150-158 (1980).
53. Fisher, D.A. & Klein, A.H. Thyroid development and disorders of thyroid function in the newborn. *N Engl J Med* **304**, 702-712 (1981).
54. Klein, A.H., Oddie, T.H., Parslow, M., Foley, T.P., Jr. & Fisher, D.A. Developmental changes in pituitary-thyroid function in the human fetus and newborn. *Early Hum Dev* **6**, 321-330 (1982).

55. Sava, L., Delange, F., Belfiore, A., Purrello, F. & Vigneri, R. Transient impairment of thyroid function in newborn from an area of endemic goiter. *J Clin Endocrinol Metab* **59**, 90-95 (1984).
56. Sunartini & Nakamura, H. Thyroid function in newborn infants from goitrous and non goitrous mothers. *Kobe J Med Sci* **37**, 265-271 (1991).
57. Narin, N., et al. Thyroid function tests in the newborn infants of preeclamptic women. *J Pediatr Endocrinol Metab* **12**, 69-73 (1999).
58. Eltom, A., Eltom, M., Idris, M. & Gebre-Medhin, M. Thyroid function in the newborn in relation to maternal thyroid status during labour in a mild iodine deficiency endemic area in Sudan. *Clin Endocrinol (Oxf)* **55**, 485-490 (2001).
59. Kelsh, M.A., et al. Primary congenital hypothyroidism, newborn thyroid function, and environmental perchlorate exposure among residents of a Southern California community. *J Occup Environ Med* **45**, 1116-1127 (2003).
60. Djemli, A., Van Vliet, G., Belgoudi, J., Lambert, M. & Delvin, E.E. Reference intervals for free thyroxine, total triiodothyronine, thyrotropin and thyroglobulin for Quebec newborns, children and teenagers. *Clin Biochem* **37**, 328-330 (2004).
61. Herbstman, J., Apelberg, B.J., Witter, F.R., Panny, S. & Goldman, L.R. Maternal, infant, and delivery factors associated with neonatal thyroid hormone status. *Thyroid* **18**, 67-76 (2008).
62. Lem, A.J., et al. Serum thyroid hormone levels in healthy children from birth to adulthood and in short children born small for gestational age. *J Clin Endocrinol Metab* **97**, 3170-3178 (2012).
63. Mutlu, M., et al. Reference intervals for thyrotropin and thyroid hormones and ultrasonographic thyroid volume during the neonatal period. *J Matern Fetal Neonatal Med* **25**, 120-124 (2012).
64. Sheikhabaei, S., Mahdavian, B., Abdollahi, A. & Nayeri, F. Serum thyroid stimulating hormone, total and free T4 during the neonatal period: Establishing regional reference intervals. *Indian J Endocrinol Metab* **18**, 39-43 (2014).
65. Jayasuriya, M.S., et al. Reference intervals for neonatal thyroid function tests in the first 7 days of life. *J Pediatr Endocrinol Metab* **31**, 1113-1116 (2018).
66. Kapelari, K., et al. Pediatric reference intervals for thyroid hormone levels from birth to adulthood: a retrospective study. *BMC Endocr Disord* **8**, 15 (2008).
67. Rastogi, M.V. & LaFranchi, S.H. Congenital hypothyroidism. *Orphanet J Rare Dis* **5**, 17 (2010).
68. Jain, V., Agarwal, R., Deorari, A.K. & Paul, V.K. Congenital hypothyroidism. *Indian J Pediatr* **75**, 363-367 (2008).
69. Demers, L.M. & Spencer, C.A. Laboratory medicine practice guidelines: laboratory support for the diagnosis and monitoring of thyroid disease. *Clin Endocrinol (Oxf)* **58**, 138-140 (2003).
70. Baloch, Z., et al. Laboratory medicine practice guidelines. Laboratory support for the diagnosis and monitoring of thyroid disease. *Thyroid* **13**, 3-126 (2003).
71. Mercado, M., Yu, V.Y., Francis, I., Szymonowicz, W. & Gold, H. Thyroid function in very preterm infants. *Early Hum Dev* **16**, 131-141 (1988).
72. Adams, L.M., Emery, J.R., Clark, S.J., Carlton, E.I. & Nelson, J.C. Reference ranges for newer thyroid function tests in premature infants. *J Pediatr* **126**, 122-127 (1995).
73. Clark, S.J., et al. Reference ranges for thyroid function tests in premature infants beyond the first week of life. *J Perinatol* **21**, 531-536 (2001).
74. Faix, J.D., Rosen, H.N. & Velazquez, F.R. Indirect estimation of thyroid hormone-binding proteins to calculate free thyroxine index: comparison of nonisotopic methods that use labeled thyroxine ("T-uptake"). *Clin. Chem* **41**, 41-47 (1995).
75. Frank, J.E., et al. Thyroid function in very low birth weight infants: effects on neonatal hypothyroidism screening. *J Pediatr* **128**, 548-554 (1996).
76. Klein, R.Z., et al. Thyroid function in very low birth weight infants. *Clin Endocrinol (Oxf)* **47**, 411-417 (1997).
77. van Wassenaer, A.G., Kok, J.H., Dekker, F.W. & de Vijlder, J.J. Thyroid function in very preterm infants: influences of gestational age and disease. *Pediatr Res* **42**, 604-609 (1997).
78. Rooman, R.P., et al. Low thyroxinaemia occurs in the majority of very preterm newborns. *Eur J Pediatr* **155**, 211-215 (1996).
79. Cuestas, R.A. & Engel, R.R. Thyroid function in preterm infants with respiratory distress syndrome. *J Pediatr* **94**, 643-646 (1979).
80. Cuestas, R.A., Lindall, A. & Engel, R.R. Low thyroid hormones and respiratory-distress syndrome of the newborn. Studies on cord blood. *N Engl J Med* **295**, 297-302 (1976).
81. Nelson, J.C. & Tomei, R.T. Dependence of the thyroxin/thyroxin-binding globulin (TBG) ratio and the free thyroxin index on TBG concentrations. *Clin Chem* **35**, 541-544 (1989).
82. Nelson, J.C. & Wilcox, R.B. Further studies on thyroxin-binding globulin-dependence in equilibrium dialysis assays of free thyroxin. *Clin Chem* **37**, 128-130 (1991).

83. Nelson, J.C., Wilcox, R.B. & Pandian, M.R. Dependence of free thyroxine estimates obtained with equilibrium tracer dialysis on the concentration of thyroxine-binding globulin. *Clin Chem* **38**, 1294-1300 (1992).
84. Dominguez, C.E., Laughlin, G.A., Nelson, J.C. & Yen, S.S. Altered binding of serum thyroid hormone to thyroxine-binding globulin in women with functional hypothalamic amenorrhea. *Fertil Steril* **68**, 992-996 (1997).
85. Fritz, K.S., Wilcox, R.B. & Nelson, J.C. Quantifying spurious free T4 results attributable to thyroxine-binding proteins in serum dialysates and ultrafiltrates. *Clin Chem* **53**, 985-988 (2007).
86. International Federation of Clinical, C., et al. Proposal of a candidate international conventional reference measurement procedure for free thyroxine in serum. *Clin Chem Lab Med* **45**, 934-936 (2007).
87. Job, L., et al. Serum free thyroxine concentration is not reduced in premature infants with respiratory distress syndrome. *J Pediatr* **131**, 489-492 (1997).
88. Nelson, J.C., Clark, S.J., Borut, D.L., Tomei, R.T. & Carlton, E.I. Age-related changes in serum free thyroxine during childhood and adolescence. *J Pediatr* **123**, 899-905 (1993).
89. Nelson, J.C., Nayak, S.S. & Wilcox, R.B. Variable underestimates by serum free thyroxine (T4) immunoassays of free T4 concentrations in simple solutions. *J Clin Endocrinol Metab* **79**, 1373-1375 (1994).
90. Nelson, J.C. & Weiss, R.M. The effect of serum dilution on free thyroxine (T4) concentration in the low T4 syndrome of nonthyroidal illness. *J Clin Endocrinol Metab* **61**, 239-246 (1985).
91. Nelson, J.C., Weiss, R.M. & Wilcox, R.B. Underestimates of serum free thyroxine (T4) concentrations by free T4 immunoassays. *J Clin Endocrinol Metab* **79**, 76-79 (1994).
92. Nelson, R.W., Ihle, S.L., Feldman, E.C. & Bottoms, G.D. Serum free thyroxine concentration in healthy dogs, dogs with hypothyroidism, and euthyroid dogs with concurrent illness. *J Am Vet Med Assoc* **198**, 1401-1407 (1991).
93. Schachter, S., et al. Comparison of serum-free thyroxine concentrations determined by standard equilibrium dialysis, modified equilibrium dialysis, and 5 radioimmunoassays in dogs. *J Vet Intern Med* **18**, 259-264 (2004).
94. Wang, R., Nelson, J.C., Weiss, R.M. & Wilcox, R.B. Accuracy of free thyroxine measurements across natural ranges of thyroxine binding to serum proteins. *Thyroid* **10**, 31-39 (2000).
95. Wilcox, R.B., Nelson, J.C. & Tomei, R.T. Heterogeneity in affinities of serum proteins for thyroxine among patients with non-thyroidal illness as indicated by the serum free thyroxine response to serum dilution. *Eur J Endocrinol* **131**, 9-13 (1994).
96. Chen, X., Zhou, Y., Zhou, M., Yin, Q. & Wang, S. Diagnostic Values of Free Triiodothyronine and Free Thyroxine and the Ratio of Free Triiodothyronine to Free Thyroxine in Thyrotoxicosis. *Int J Endocrinol* **2018**, 4836736 (2018).
97. Jonklaas, J., et al. Total and free thyroxine and triiodothyronine: measurement discrepancies, particularly in inpatients. *Clin Biochem* **47**, 1272-1278 (2014).
98. Nelson, J.C., Wang, R., Asher, D.T. & Wilcox, R.B. The nature of analogue-based free thyroxine estimates. *Thyroid* **14**, 1030-1036 (2004).
99. Steele, B.W., et al. Total long-term within-laboratory precision of cortisol, ferritin, thyroxine, free thyroxine, and thyroid-stimulating hormone assays based on a College of American Pathologists fresh frozen serum study: do available methods meet medical needs for precision? *Arch Pathol Lab Med* **129**, 318-322 (2005).
100. Wang, D., et al. Reference intervals for thyroid-stimulating hormone, free thyroxine, and free triiodothyronine in elderly Chinese persons. *Clin Chem Lab Med* **57**, 1044-1052 (2019).
101. Wang, X., Chen, H., Lin, J.M. & Ying, X. Development of a highly sensitive and selective microplate chemiluminescence enzyme immunoassay for the determination of free thyroxine in human serum. *Int J Biol Sci* **3**, 274-280 (2007).
102. Wang, Y., Zhang, Y.X., Zhou, Y.L. & Xia, J. Establishment of reference intervals for serum thyroid-stimulating hormone, free and total thyroxine, and free and total triiodothyronine for the Beckman Coulter DxI-800 analyzers by indirect method using data obtained from Chinese population in Zhejiang Province, China. *J Clin Lab Anal* **31**(2017).
103. Zhang, J., et al. Establishment of trimester-specific thyroid stimulating hormone and free thyroxine reference interval in pregnant Chinese women using the Beckman Coulter UniCel DxI 600. *Clin Chem Lab Med* **53**, 1409-1414 (2015).
104. Klein, A.H., Foley, B., Kenny, F.M. & Fisher, D.A. Thyroid hormone and thyrotropin responses to parturition in premature infants with and without the respiratory distress syndrome. *Pediatrics* **63**, 380-385 (1979).
105. López, M., Alvarez, C.V., Nogueiras, R. & Diéguez, C. Energy balance regulation by thyroid hormones at central level. *Trends Mol. Med* **19**, 418-427 (2013).

106. Warner, A. & Mittag, J. Thyroid hormone and the central control of homeostasis. *J. Mol. Endocrinol* **49**, R29-R35 (2012).
107. Mullur, R., Liu, Y.Y. & Brent, G.A. Thyroid hormone regulation of metabolism. *Physiol Rev* **94**, 355-382 (2014).
108. Mendoza, A. & Hollenberg, A.N. New insights into thyroid hormone action. *Pharmacol Ther* **173**, 135-145 (2017).
109. Pinheiro, J.B., DM. *Mixed-effects models in S and S-PLUS*, (Springer, 2000).
110. Jiang, J. *Linear and generalized linear mixed models and their applications*, (Springer, 2007).
111. Lubkowska, A., Szymanski, S. & Chudecka, M. Surface Body Temperature of Full-Term Healthy Newborns Immediately after Birth-Pilot Study. *Int J Environ Res Public Health* **16**(2019).
112. Enerback, S. Human brown adipose tissue. *Cell Metab* **11**, 248-252 (2010).
113. Lidell, M.E. Brown Adipose Tissue in Human Infants. *Handb Exp Pharmacol* **251**, 107-123 (2019).
114. Scheele, C. & Wolfrum, C. Brown Adipose Crosstalk in Tissue Plasticity and Human Metabolism. *Endocr Rev* **41**, 53-65 (2020).
115. Hey, E.N., Katz, G. & O'Connell, B. The total thermal insulation of the new-born baby. *J Physiol* **207**, 683-698 (1970).
116. Wheldon, A.E. & Harpin, V.A. Metabolic rate in newborn babies in thermoneutral conditions and when overheated. *Early Hum Dev* **6**, 249-252 (1982).
117. Davidson, S., Reina, N., Shefi, O., Hai-Tov, U. & Akselrod, S. Spectral analysis of heart rate fluctuations and optimum thermal management for low birth weight infants. *Med Biol Eng Comput* **35**, 619-625 (1997).
118. Thomas, K.A. Infant weight and gestational age effects on thermoneutrality in the home environment. *J Obstet Gynecol Neonatal Nurs* **32**, 745-752 (2003).
119. Laptook, A.R. & Watkinson, M. Temperature management in the delivery room. *Semin Fetal Neonatal Med* **13**, 383-391 (2008).
120. Organization, W.H. Thermal Protection of the Newborn: a practical guide. in *Maternal and Newborn Health* (ed. Organization, W.H.) (Geneva, 1997).
121. Smales, O.R. & Kime, R. Thermoregulation in babies immediately after birth. *Arch Dis Child* **53**, 58-61 (1978).
122. Smales, O.R. & Hull, D. Metabolic response to cold in the newborn. *Arch Dis Child* **53**, 407-410 (1978).
123. Bach, V., Telliez, F., Krim, G. & Libert, J.P. Body temperature regulation in the newborn infant: interaction with sleep and clinical implications. *Neurophysiol Clin* **26**, 379-402 (1996).
124. Silverman, W.A., Fertig, J.W. & Berger, A.P. The influence of the thermal environment upon the survival of newly born premature infants. *Pediatrics* **22**, 876-886 (1958).
125. Hey, E.N. & Katz, G. Temporary loss of a metabolic response to cold stress in infants of low birthweight. *Arch Dis Child* **44**, 323-330 (1969).
126. Lunze, K., Bloom, D.E., Jamison, D.T. & Hamer, D.H. The global burden of neonatal hypothermia: systematic review of a major challenge for newborn survival. *BMC Med* **11**, 24 (2013).
127. Trevisanuto, D., Testoni, D. & de Almeida, M.F.B. Maintaining normothermia: Why and how? *Semin Fetal Neonatal Med* **23**, 333-339 (2018).
128. Himms-Hagen, J. Does thermoregulatory feeding occur in newborn infants? A novel view of the role of brown adipose tissue thermogenesis in control of food intake. *Obes Res* **3**, 361-369 (1995).
129. Hulbert, A.J. & Else, P.L. Basal metabolic rate: history, composition, regulation, and usefulness. *Physiol Biochem. Zool* **77**, 869-876 (2004).
130. Cannon, B. & Nedergaard, J. Brown adipose tissue: function and physiological significance. *Physiol Rev* **84**, 277-359 (2004).
131. Perlman, J. & Kjaer, K. Neonatal and Maternal Temperature Regulation During and After Delivery. *Anesth Analg* **123**, 168-172 (2016).
132. Sarasniemi, J.T., et al. Skin temperature may not yield human brown adipose tissue activity in diverse populations. *Acta Physiol (Oxf)* **224**, e13095 (2018).
133. Jang, C., et al. Infrared thermography in the detection of brown adipose tissue in humans. *Physiol Rep* **2**(2014).
134. Chondronikola, M., Beeman, S.C. & Wahl, R.L. Non-invasive methods for the assessment of brown adipose tissue in humans. *J Physiol* **596**, 363-378 (2018).
135. Heimann, K., et al. Infrared thermography for detailed registration of thermoregulation in premature infants. *J Perinat Med* **41**, 613-620 (2013).
136. Knobel, R.B., Guenther, B.D. & Rice, H.E. Thermoregulation and thermography in neonatal physiology and disease. *Biol Res Nurs* **13**, 274-282 (2011).
137. Gatidis, S., et al. Is It Possible to Detect Activated Brown Adipose Tissue in Humans Using Single-Time-Point Infrared Thermography under Thermoneutral Conditions? Impact of BMI and Subcutaneous Adipose Tissue Thickness. *PLoS One* **11**, e0151152 (2016).

138. Koprowski, R. & Olczyk, P. Segmentation in dermatological hyperspectral images: dedicated methods. *Biomed Eng Online* **15**, 97 (2016).
139. Owda, A.Y., et al. Millimeter-wave emissivity as a metric for the non-contact diagnosis of human skin conditions. *Bioelectromagnetics* **38**, 559-569 (2017).
140. Togawa, T. & Saito, H. Non-contact imaging of thermal properties of the skin. *Physiol Meas* **15**, 291-298 (1994).
141. Huang, J. & Togawa, T. Improvement of imaging of skin thermal properties by successive thermographic measurements at a stepwise change in ambient radiation temperature. *Physiol Meas* **16**, 295-301 (1995).
142. Sanchez-Marin, F.J., Calixto-Carrera, S. & Villasenor-Mora, C. Novel approach to assess the emissivity of the human skin. *J Biomed Opt* **14**, 024006 (2009).
143. Mitchell, D., Wyndham, C.H. & Hodgson, T. Emissivity and transmittance of excised human skin in its thermal emission wave band. *J Appl Physiol* **23**, 390-394 (1967).
144. Mitchell, D., Wyndham, C.H., Hodgson, T. & Nabarro, F.R. Measurement of the total normal emissivity of skin without the need for measuring skin temperature. *Phys Med Biol* **12**, 359-366 (1967).
145. Watmough, D.J. & Oliver, R. Emissivity of human skin in the waveband between 2micra and 6micra. *Nature* **219**, 622-624 (1968).
146. Watmough, D.J. & Oliver, R. Wavelength dependence of skin emissivity. *Phys Med Biol* **14**, 201-204 (1969).
147. Steketee, J. Spectral emissivity of skin and pericardium. *Phys Med Biol* **18**, 686-694 (1973).
148. Togawa, T. Non-contact skin emissivity: measurement from reflectance using step change in ambient radiation temperature. *Clin Phys Physiol Meas* **10**, 39-48 (1989).
149. Boylan, A., Martin, C.J. & Gardner, G.G. Infrared emissivity of burn wounds. *Clin Phys Physiol Meas* **13**, 125-127 (1992).
150. Huang, J. & Togawa, T. Measurement of the thermal inertia of the skin using successive thermograms taken at a stepwise change in ambient radiation temperature. *Physiol Meas* **16**, 213-225 (1995).
151. Bernard, V., Staffa, E., Mornstein, V. & Bourek, A. Infrared camera assessment of skin surface temperature--effect of emissivity. *Phys Med* **29**, 583-591 (2013).
152. Abbas, A.K. & Leonhardt, S. Intelligent neonatal monitoring based on a virtual thermal sensor. *BMC Med Imaging* **14**, 9 (2014).
153. Keenan, E., Gethin, G., Flynn, L., Watterson, D. & O'Connor, G.M. Enhanced thermal imaging of wound tissue for better clinical decision making. *Physiol Meas* **38**, 1104-1115 (2017).
154. Gatt, A., et al. The identification of higher forefoot temperatures associated with peripheral arterial disease in type 2 diabetes mellitus as detected by thermography. *Prim Care Diabetes* **12**, 312-318 (2018).
155. Glavas, H., Vukobratovic, M. & Keser, T. Infrared thermography as control of handheld IPL device for home-use. *J Cosmet Laser Ther* **20**, 269-277 (2018).
156. Goulart, C., Valadao, C., Delisle-Rodriguez, D., Caldeira, E. & Bastos, T. Emotion analysis in children through facial emissivity of infrared thermal imaging. *PLoS One* **14**, e0212928 (2019).
157. Law, J., Morris, D.E., Budge, H. & Symonds, M.E. Infrared Thermography. *Handb Exp Pharmacol* **251**, 259-282 (2019).
158. Fernández-Cuevas, I., et al. Classification of factors influencing the use of infrared thermography in humans: A review. *Infrared Physics & Technology* **71**, 28-55 (2015).
159. Eichenfield, L., I, F., E, M. & A, Z. *Neonatal and Infant Dermatology*, (Elsevier, Unites Stated of America, 2014).
160. Knobel-Dail, R.B., Holditch-Davis, D., Sloane, R., Guenther, B.D. & Katz, L.M. Body temperature in premature infants during the first week of life: Exploration using infrared thermal imaging. *J Therm Biol* **69**, 118-123 (2017).
161. Abbas, A.K., Heimann, K., Blazek, V., Orlikowsky, T. & Leonhardt, S. Neonatal infrared thermography imaging: Analysis of heat flux during different clinical scenarios. *Infrared Physics & Technology* **55**, 538-548 (2012).
162. Abbas, A.K., Jergus, K., Heiman, K., Orlikowsky, T. & Leonhardt, S. Neonatal Infrared Thermography Monitoring. in *Neonatal Monitoring Technologies: Design for Integrated Solutions* (eds. Chen, W., Bambang Oetomo, S. & Feijs, L.) (United States of America, 2012).

Reviewer #4 (Remarks to the Author):

As mentioned previously, the manuscript describes a protocol for quantifying BAT thermogenesis in neonates, which is highly valuable to researchers investigating changes in adipose tissue and metabolism across development. The study documents interesting associations between BAT thermogenesis and bone morphogenetic protein 8B (BMP8B). The authors have adequately addressed all of my previous comments.

REVIEWER#4

General comment: *As mentioned previously, the manuscript describes a protocol for quantifying BAT thermogenesis in neonates, which is highly valuable to researchers investigating changes in adipose tissue and metabolism across development. The study documents interesting associations between BAT thermogenesis and bone morphogenetic protein 8B (BMP8B). The authors have adequately addressed all of my previous comments.*

Response: We thank very much the Reviewer for the positive view of our manuscript and his/her words on our work. We also thank again the Reviewer for his/her comments which highly improved the quality of our study.